



# Simulating asymmetric growth and retreat of Hardangerjøkulen ice cap in southern Norway since the mid-Holocene

Henning Åkesson[1], Kerim H. Nisancioglu[1,2], Rianne H. Giesen[3], and Mathieu Morlighem[4]

[1]Department of Earth Science, University of Bergen and Bjerknes Centre for Climate Research, Allégaten 70, 5007 Bergen, Norway
[2]Centre for Earth Evolution and Dynamics, Postbox 1028 Blindern, University of Oslo, 0315 Oslo, Norway
[3]Institute for Marine and Atmospheric research, Utrecht University, Utrecht, The Netherlands
[4]University of California, Irvine, Department of Earth System Science, 3218 Croul Hall, Irvine, CA 92697-3100, USA

*Correspondence to:* Henning Åkesson (henning.akesson@uib.no)

**Abstract.** Changes to the volume of glaciers and ice caps currently amount to half of the total cryospheric contribution to sea-level rise and are projected to remain substantial throughout the 21st century. To simulate glacier behavior on centennial and longer time scales, models rely on simplified dynamics and tunable parameters for processes not well understood. Model calibration is often done using present-day observations, even though the relationship between parameters and parametrized

processes may be altered for significantly different glacier states.

In this study, we simulate the evolution of the Hardangerjøkulen ice cap in southern Norway from the mid-Holocene through the Little Ice Age (LIA) to the present-day. For both the calibration and transient experiments, we run an ensemble using a two-dimensional ice flow model with local mesh refinement. For the Holocene, we apply a simple surface mass balance forcing based on climate reconstructions. For the LIA until 1962, we use geomorphological evidence and measured outlet

glacier positions to find a mass balance history, while from 1963 until today we use direct mass balance measurements.

Given a linear climate forcing, we find that Hardangerjøkulen grew from ice-free conditions in the mid-Holocene, to its maximum LIA extent in a highly non-linear fashion. During the fastest stage of growth (2200–1200 BP), the ice cap tripled its ice volume over only 1000 years. We also reveal an intriguing spatial asymmetry during advance and retreat; the western ice cap and the northern outlet glacier Midtdalsbreen grow first and disappear first. In contrast, the eastern part, including the

northeastern outlet glacier Blåisen, grows last and disappears last.

Furthermore, volume and area of several outlet glaciers, as well as of the entire ice cap, vary out-of-phase for multiple centuries during the late Holocene, before varying in-phase approaching the LIA. We relate this to bed topography and the mass balance-altitude feedback, and challenge canonical linear assumptions between ice cap extent and glacier proxy records. Thus, we provide new insight into long-term dynamical response of ice caps to climate change, relevant for paleoglaciological

studies and future predictions.

Our model simulates ice cap extent and outlet glacier length changes from the LIA until today that are close to observations. We show that present-day Hardangerjøkulen is extremely sensitive to surface mass balance changes, mainly due to a strong mass balance-altitude feedback for the gently sloping surface topography of the ice cap.





# 1 Introduction

Recent global decline in glacier ice across all continents (Gardner et al., 2013) is viewed as one of the clearest signs of the impacts of a warming climate (Vaughan et al., 2013). Further projected accelerated mass loss (Radic et al., 2013; Giesen and Oerlemans, 2013) is predicted to have considerable impact on 21st century sea level rise (Marzeion et al., 2012; Church et al., 2013) and beyond (Levermann et al., 2013). Socioeconomic effects include severe changes to hydropower operations, agriculture, tourism and ecosystems.

Given the inertia of ice sheets, assessments of the response of glaciers and small ice caps (GICs) to climate are essential. Although the 170 000 GICs in the world are relatively small compared to the Greenland and Antarctic ice sheets, they constitute about half of the current cryospheric contribution to sea level rise (Shepherd et al., 2012; Vaughan et al., 2013), a distribution projected to remain similar throughout the 21st century (Church et al., 2013).

Accurate predictions of the future response of GICs rely on our understanding of the relative importance of ice dynamics and surface mass balance (hereafter referred to as 'mass balance'). For comparison with current rapid changes and future predictions, glacier and climate reconstructions (e.g. Karlén, 1976; Nesje et al., 1991; Bakke et al., 2005) provide an invaluable baseline. Glacier reconstructions build on glaciological assumptions, including that sediment input is a function of glacier (erosive) area and mass balance throughput (e.g. Hallet et al., 1996). Using models in combination with independent glacier reconstructions has the potential to deepen our understanding of glacier-climate interactions and give insight into the physics operating on these time scales.

Recent interest in so called Full Stokes ice dynamical models (e.g. Larour et al., 2012; Gagliardini et al., 2013), accounting for the complete stress balance associated with ice sheet and glacier dynamics has led to valuable progress in process understanding. Nonetheless, simpler models are generally preferred, as long as they can accurately describe ice flow and climatic response on the time scales of interest. Due to their much lower computational cost, simple ice dynamical models allow for extensive 'ensemble' experiments, assessing the effect of a suite of parameters on model outcome (Alley and Joughin, 2012).

In this study, we present simulations of the Hardangerjøkulen ice cap in southern Norway since 4000 BP, using a simple ice dynamical model. Several GIC modeling studies have investigated glacier response on time scales over a few centuries (e.g Giesen and Oerlemans, 2010; Adalgeirsdóttir et al., 2011; Zekollari et al., 2014) as well as millennia (e.g. Flowers et al., 2008; Laumann and Nesje, 2014). Studies focusing on the glacier evolution since the Little Ice Age (LIA) normally perturb a present-day glacier with a climate anomaly relative to a modern climatology to serve as initial conditions. Here, we instead present a plausible ice cap history over several thousand years before the LIA, which is then used as a starting point for LIA to present-day simulations. In addition, we carry out an extensive evaluation of the sensitivity to dynamical model parameters, which in most other studies is restricted to the calibration phase, and in a sense 'lost' when transient model simulations start.

Using the above methodology, we aim to advance our understanding of long-term (centuries to millennia) dynamic aspects of glacier change, and to quantify the sensitivity of Hardangerjøkulen ice cap to climatic change. We also show that glacier reconstructions can be improved by considering the underlying bed topography and interacting ice dynamics.




This paper is organized as follows. First, the Hardangerjøkulen ice cap is described in Sect. 2, after which our model strategy and calibration is outlined in Sect. 3. Results from the mid- to late Holocene simulations, as well as from the LIA until today, are presented in Sect. 4. We analyze the sensitivity to model parameters, mass balance and ice dynamics in Sect. 5, including the implications for our modelled Holocene evolution. Finally, we discuss the sensitivity of present-day Hardangerjøkulen to

future changes in mass balance.

## 2 Hardangerjøkulen ice cap

### 2.1 Geometry

#### 2.1.1 Surface topography

Hardangerjøkulen ice cap (60°55'N, 7°25'E) has a present-day area of 73 km$^2$ (Andreassen et al., 2012) and is located at

the western flank of the Hardangervidda mountain plateau. The ice cap is rather flat in the interior with steeper glaciers draining the plateau (Fig. 1). The largest outlet glaciers are Rembesdalskåka (facing W-SW; 17.4 km$^2$), Midtdalsbreen (NE; 6.8 km$^2$), Blåisen (NE; 6.6 km$^2$) and Vestre Leirbotnskåka (S-SE; 8 km$^2$). Surface elevation ranges from 1865 to 1020 m a.s.l. (Andreassen et al., 2015), with 80 % of the ice cap area, and 70 % of Rembesdalskåka, situated above the mean equilibrium-line altitude (ELA) at 1640 m a.s.l. (1963-2007 average; Giesen, 2009). Rembesdalskåka drains towards the dammed lake

Rembesdalsvatnet, located ∼1 km from the present-day glacier terminus (Kjøllmoen et al., 2011).

#### 2.1.2 Ice thickness and bed topography

A number of surveys have mapped the ice thickness at Hardangerjøkulen (e.g. Sellevold and Kloster, 1964; Elvehøy et al., 1997; Østen, 1998, K. Melvold, unpubl. data), with the highest measurement density for Midtdalsbreen (Fig. 2.12a in Giesen, 2009).

A combination of automatic and manual interpolation and extrapolation was used to produce the final ice thickness map (Elvehøy et al., 1997; Willis et al., 2012), and thereby a map of bed topography (Fig. 1). In areas with dense measurements, ice thickness was interpolated using methods detailed in Melvold and Schuler (2008). In sparsely measured areas, ice thickness $H$ was estimated directly from the surface slope $\alpha$, assuming perfect plasticity (Paterson, 1994, p. 240):

$$H = \frac{\tau_0}{\rho_i g \nabla s},$$ (1)

where $\tau_0$ is the yield stress, $\rho_i$ ice density, $g$ gravitational acceleration and $\nabla s$ the surface slope. Based on detailed ice thickness measurements and knowledge of the surface slope on Midtdalsbreen, a yield stress of 150-180 kPa was used, in agreement with other mountain glaciers (Cuffey and Paterson, 2010, p.297; Zekollari et al., 2013). Some manual interpolation was required in areas with small surface slopes (i.e. at ice divides and ice ridges), as well as near ice margins, to obtain a continuous decrease in ice thickness (K. Melvold, pers. comm.).



### 2.1.3 Outlet glacier changes since the Little Ice Age

The 'Little Ice Age' (LIA) maximum for Midtdalsbreen is dated to 1750 AD with lichenometry (Andersen and Sollid, 1971). For Rembesdalskåka, the outermost terminal moraine has not been dated, but is assumed to originate from the LIA maximum.

Front observations for Rembesdalskåka began in 1917, have subsequently been performed during several periods, and are
since 1995 done annually. For Midtdalsbreen, an annual length change record exists from 1982 onwards (Kjøllmoen et al., 2011). Rembesdalskåka has retreated almost 2 km from its LIA maximum extent and Midtdalsbreen ∼1 km.

Both outlet glaciers advanced in response to snowy winters around 1990. The terminus change from 1988 to 2000 for Rembesdalskåka was +147 m and for Midtdalsbreen +46 m. By 2013, the two glaciers had retreated 332 m and 164 m from their positions in 2000, respectively (Andreassen et al., 2005; Kjøllmoen et al., 2011; Cryoclim.net, 2014).

Length changes extracted from maps and satellite imagery, moraine positions and direct front measurements are combined to derive length records for the two outlet glaciers for the period 1750-2008. For Rembesdalskåka, we use the same flowline as NVE uses for their mass balance measurements (H. Elvehøy, pers. comm.). The NVE flowline for Midtdalsbreen was slightly modified to better correspond with the maximum ice velocities. Since changes are only made upglacier of the present-day margin, they do not interfere with the area where data of frontal changes exist.

### 15 2.1.4 Holocene changes

Based on lake sediments and terrestrial deposits, Hardangerjøkulen is estimated to have been practically absent from c. 7500 to 4800 BP (Dahl and Nesje, 1994), though a short-lived glacier advance is documented for the southern side of the ice cap at c. 7000 BP (Nesje et al., 1994). Some high-frequency glacier fluctuations of local northern glaciers occurred during the period 4800-3800 BP, after which Hardangerjøkulen has been present continuously (Dahl and Nesje, 1994).

### 20 2.2 Climate and mass balance

#### 2.2.1 Present climate

Southern Norway is located in the Northern Hemisphere westerly wind belt and is heavily influenced by moist, warm air picked up by the frequent storms coming off the Atlantic Ocean (Uvo, 2003). When these winds reach the mountainous west coast, orographic lifting occurs and precipitation falls as rain or snow, depending on elevation. Conversely, eastern Norway is located
in the rain shadow of the coastal mountains and the high mountain plateau Hardangervidda.

This west-east precipitation gradient is illustrated by the mean annual precipitation for 1961-1990 over southern Norway. Precipitation in Bergen, 65 km west of Hardangerjøkulen, reaches 2250 mm a$^{-1}$. In contrast, Oslo in eastern Norway receives 763 mm precipitation per year. Liset, 17 km southeast of the summit of Hardangerjøkulen receives 1110 mm a$^{-1}$, while Finse, 8 km northeast of the summit, experiences 1030 mm a$^{-1}$ and has a mean annual temperature of -2.1°C (data from eklima.no,
Norwegian Meteorological Institute).





### 2.2.2 Holocene and Little Ice Age climate

Reconstructions for southern Norway based on pollen and chironomids suggest that summer temperatures were up to 2°C higher than present in the period between 8000–4000 BP, when solar insolation was higher (Nesje and Dahl, 1991; Bjune et al., 2005; Velle et al., 2005a). At c. 4000 BP, when reappearance of northern Hardangerjøkulen is documented (Dahl and Nesje, 1994), temperatures at Finse had decreased to about 0.5 °C below present-day (Velle et al., 2005b).

Precipitation changes are less well-known, but the mid-Holocene in southern Scandinavia was likely drier than present (Dahl and Nesje, 1996; Seppä et al., 2005), implying unfavourable conditions for glaciers.

The period from mid-Holocene to the LIA was probably characterized by a gradual warming and wetting trend (Dahl and Nesje, 1996), wherein there may have been a warming event lasting for several centuries around 2000 BP (Velle et al., 2005a).

The LIA climate in southern Norway likely had more precipitation (Nesje and Dahl, 2003; Nesje et al., 2008; Rasmussen et al., 2010) and was c. 0.5-1.0 °C colder than present (Kalela-Brundin, 1999; Nordli et al., 2003), though some reconstructions indicate milder summers during the first quarter of the 18th century (Kalela-Brundin, 1999).

### 2.2.3 Surface mass balance

Glaciological mass balance measurements started on Rembesdalskåka in 1963. The mean net balance for the period 1963–2010 was slightly positive (+0.08 m water equivalent (w.e.)), divided into a winter balance of +2.10 m w.e. and a summer balance of -2.03 m w.e. (Kjøllmoen et al., 2011).

For Midtdalsbreen, there are only mass balance measurements for 2000-2001 (Krantz, 2002). This two-year time series is too short for a robust surface mass balance comparison between the two outlet glaciers.

Specific mass balance gradients for the entire elevation range of Rembesdalskåka exist for 35 of the 45 mass balance years in the period 1963–2007. The interannual variability around the mean winter profile is similar at all elevations, while the range in summer balances increases from high to low elevations (see Fig. 2.7a in Giesen (2009)). The decrease in (mainly winter) mass balance at the highest elevations can probably be explained by snow redistribution by wind, but to the authors' knowledge, no studies have quantified this effect on the mass balance at Hardangerjøkulen.

The net balance gradient has a similar shape for most years, and the relation between net mass balance and altitude is approximately linear from the terminus up to 1675 m a.s.l. (Fig. 2), with a mass balance gradient of 0.0097 m w.e. per m altitude. The net mass balance is zero at 1640 m a.s.l., marking the equilibrium-line altitude (ELA). Above the ELA, the mass balance gradient decreases with altitude, approximated in Fig. 2 by a second-order polynomial.

### 2.3 Ice dynamics

### 2.3.1 Basal conditions

Although bed conditions are not well-known, based on the sparse sediment cover in the surrounding areas (Andersen and Sollid, 1971), we assume Hardangerjøkulen to be hard-bedded, i.e. without any deformable subglacial sediments present. Given its



climatic setting and judging from the radar investigations mentioned in Sect. 2.1.2, Hardangerjøkulen can be characterized as a temperate ice cap. Based on temperature measurements, Midtdalsbreen has been suggested to be cold-based in its lowermost parts (Hagen, 1978; Konnestad, 1996; Reinardy et al., 2013). We however expect that this has a minor effect on ice flow for Midtdalsbreen and Hardangerjøkulen as a whole.

### 2.3.2 Surface velocities

For the lower ablation zone of Midtdalsbreen, surface speeds of 4–40 m a$^{-1}$ were measured during summer 2000 (Vaksdal, 2001). In addition, ice velocities was derived from Global Positioning System (GPS) unit recordings at nine locations on Hardangerjøkulen during the period May 2005–September 2007 (Giesen, 2009). One GPS was mounted on the automatic weather station (AWS) on Midtdalsbreen, the other eight were situated on stakes at the ELA of the main outlet glaciers (Fig. 1). These data show highest velocities for the largest outlet glacier Rembesdalskåka (46 m a$^{-1}$). Velocities at Midtdalsbreen (33 m a$^{-1}$ at the ELA and ∼20–22 m a$^{-1}$ at the AWS) are within the range of ablation zone summer velocities suggested by Vaksdal (2001).

Since velocities have only been measured for single years or shorter, these observations provide guidance rather than serving as calibration or validation data for our model. To the authors' knowledge, there are no velocity data derived from remote sensing platforms.

## 3 Model description and setup

### 3.1 Ice flow model

We use the Ice Sheet System Model (ISSM; Larour et al., 2012), a finite element ice flow model primarily developed for high-resolution, higher-order modeling of ice sheets using a parallel software architecture. There are many capabilities and modules in ISSM; only those relevant for this paper are covered here. For a complete description, including a more comprehensive section on model numerics and architecture, we refer to Larour et al. (2012) and http://issm.jpl.nasa.gov.

We use the two-dimensional (2d), vertically integrated Shallow Ice Approximation (SIA) within ISSM, meaning horizontal velocities correspond to averaged velocities over the ice column. The SIA is based on a scaling analysis of the full Stokes stress balance (Hutter, 1983; Morland, 1984). This scaling argument carries the assumption that typical glacier length $L$, is much larger than the typical ice thickness $H$. For this purpose, the aspect-ratio $\epsilon$ is defined as

$$\epsilon = \frac{H}{L}, \tag{2}$$

where $\epsilon$ describes the 'shallowness' of an ice mass. An aspect-ratio much smaller than unity is required for the SIA to be valid. Generally, the smaller the $\epsilon$, the more accurate the SIA is (Le Meur et al., 2004; Winkelmann et al., 2011). For Hardangerjøkulen, the characteristic horizontal scale is 4 to 8 km, and the characteristic ice thickness ∼200 m, giving an $\epsilon$ of 0.05 to 0.025, which is acceptable for using the SIA (Le Meur and Vincent, 2003).



Where bed and surface topography is complex, lateral drag and longitudinal stress gradients may become important. Still, the SIA has proven accurate in representing glacier length and volume fluctuations on decadal and longer time scales (Leysinger-Vieli and Gudmundsson, 2004).

Hardangerjøkulen has relatively gentle surface slopes and lacks areas of very fast flow. The SIA is therefore a viable option when studying this ice cap on climatic time scales. Similar studies on Hardangerjøkulen (Giesen and Oerlemans, 2010), ice caps in Iceland (Guðmundsson et al., 2009; Adalgeirsdóttir et al., 2011), and glaciers in the French Alps (Le Meur et al., 2007) indicate that employing SIA models on alpine glaciers and small ice caps on climatic time scales gives satisfactory results. Because of its simplicity, SIA is computationally efficient (Rutt et al., 2009). This enables modeling of ice cap evolution over long time scales and for a wide parameter space, both key aims of this study.

### 3.1.1 Ice deformation and sliding

The constitutive relationship relating stress $\tau$ to ice deformation (strain rate) is Glen's flow law (Glen, 1955), which for the special case of vertical shear stress only (SIA) is

$$\dot{\epsilon} = A\tau^n \tag{3}$$

where $\dot{\epsilon}$ is the strain rate tensor, $A$ is a temperature dependent flow factor accounting for ice rheology and $n = 3$ is Glen's flow law exponent. We use a spatially constant flow factor $A$, assuming homogeneous ice temperature and material properties across the ice cap.

In contrast to many other studies, where a tuned 'best-fit' parameter combination is used throughout all simulations, we perform ensemble runs for a parameter space of different flow factors and sliding parameters (described below), for both the calibration procedure and subsequent model runs.

Consistent with SIA theory, vertically averaged ice deformational velocities $\bar{u}_d$ are calculated by

$$\bar{u}_d = \frac{2AH}{n+2}\tau_d^n \tag{4}$$

Since the SIA assumes driving stress $\tau_d$ to be equal to the basal shear stress $\tau_b$, the basal shear stress $\tau_b$ can be written as

$$\boldsymbol{\tau_b} = \boldsymbol{\tau_d} = \rho_i g H \nabla s \tag{5}$$

where $\rho_i$ is the density of ice, $g$ the gravitational acceleration and $\nabla s$ the surface slope.

SIA is strictly only valid for a no-slip bed (Gudmundsson, 2003; Hindmarsh, 2004). However, Hardangerjøkulen is a temperate ice cap, and summer speed-ups have been observed at Midtdalsbreen (Willis, 1995; Willis et al., 2012), indicating basal motion.

Several previous studies (e.g. Ritz et al., 1996; Payne et al., 2000; Rutt et al., 2009) employing the SIA use a linear Weertman sliding formulation (Weertman, 1964), where basal velocities are a function of the basal shear stress, which for the SIA is equivalent to setting basal velocities $u_b$ proportional to the driving stress:

$$u_b = \beta\tau_b^m, \tag{6}$$





where $\beta$ is a (tuning) basal sliding parameter. $\beta$ can be set spatially and temporally constant, or be a function of temperature, basal water depth, basal water pressure, bed roughness or other factors, and $m$ is the sliding law exponent, which equals one for the linear sliding law we apply. The basal velocity $u_b$ is added to the deformational velocity, so that total vertically averaged velocity becomes $\bar{u} = \bar{u}_d + u_b$.

In this study, the basal sliding parameter $\beta$ is assumed spatially and temporally constant. In reality, sliding likely varies both in space and time in accordance with varying basal hydrology, bed roughness and material properties. However, we consider it too speculative to apply *ad-hoc* variations in basal sliding without proper validation. ISSM has capabilities to perform inversions for basal friction based on data assimilation techniques (e.g. MacAyeal, 1993; Morlighem et al., 2010), but this requires more extensive velocity data coverage than what is available for Hardangerjøkulen.

### 3.1.2   Mass transport

For the vertically-integrated ice flow model used in this study, the two-dimensional continuity equation states

$$\frac{\partial H}{\partial t} = -\nabla \cdot (uH) + \dot{M} \tag{7}$$

where $u$ is the vertically averaged ice velocity (m a$^{-1}$) and $\dot{M}$ the surface mass balance rate (m ice equivalent). The basal melt rate is assumed negligible, and calving is not included in the model. Rembesdalskåka likely terminated in lake Rembesdalsvatnet during the LIA and the northwestern ice cap presently terminates in water, however we expect this to have minor effect on ice dynamics.

### 3.1.3   Mesh and time stepping

Following methods outlined in Hecht (2006) and Morlighem et al. (2011), an anisotropic mesh with resolution 200-500 m was constructed using local mesh refinement based on modelled velocities for a steady-state ice cap close to observed LIA extent. This ice cap was reached using our 'best-fit' deformation and sliding parameters (Sect. 3.2.2) on a uniform mesh, and a mass balance perturbation forcing the ice cap to advance to terminus positions close to the LIA extent.

Even though our surface digital elevation model (DEM) has higher resolution than this (100 m), we choose the highest mesh resolution to be 200 m, since this is more in line with the assumptions of the SIA. It also enables us to carry out Holocene runs and our ensemble study at lower computational cost.

We use a finite difference scheme in time, where a time step of 0.02 years was found low enough to avoid numerical instabilities.

### 3.2   Experimental setup and calibration

### 3.2.1   Mass balance forcing

A vertical reference mass balance gradient $B_{ref}$ is derived from observed specific mass balance gradients, which exist for 35 of the 45 years spanning 1963–2007 (Fig. 2). Mass balance $B(z,t)$ for any point in time is calculated by shifting $B_{ref}$ by a





mass balance anomaly $\Delta B(t)$ at all elevations (Oerlemans, 1997a):

$$B(z,t) = B_{ref}(z) + \Delta B(t) \tag{8}$$

$B_{ref}$ is defined to represent zero mass balance for the present-day surface topography. The measured mean net balance for Rembesdalskåka was -0.175 m w.e., and $B_{ref}$ is therefore derived by shifting the mean observed gradient by +0.175 m w.e.
(Giesen, 2009).

A mass balance-altitude feedback is included in the model by recalculating the mass balance $B(z,t)$ at a specific point for each time step according to the updated surface elevation. The elevation of the maximum net mass balance is not adapted to changes in the ice cap summit elevation, as the effect on modelled ice volume is minor (Giesen, 2009).

### 3.2.2 Ensemble calibration of ice deformation and sliding parameters

To calibrate model parameters governing ice deformation and basal sliding, we use the 1995 surface DEM as the initial condition. This topography is based on aerial photographs taken by the Norwegian Mapping Authority (Statens Kartverk), and corresponds to the period when most of the bed topography was mapped.

We run the model with constant climate forcing, using our reference mass balance gradient (i.e. with $\Delta B(t) = 0$ in Eq. 8), until a steady-state is reached.

Since we run the model with a mass balance gradient averaged over several decades, it is important that there was no large climate-geometry imbalance for this period. Indeed, the ice cap was in close to steady-state between the early 1960s and 1995, since surface elevation change from 1961 to 1995 was $\pm$ 10 m (Andreassen and Elvehøy, 2001).

In reality, an ice cap is never in exact steady-state, but it is still a useful concept to understand model sensitivity (Adalgeirsdóttir et al., 2011). To investigate model sensitivity to deformation and sliding parameters, and to find a 'best-fit' combination
for our historic runs, we run an ensemble of 24 possible parameter combinations, well enclosed by values used in the literature. For the flow factor $A$, we investigate values from $A = 0.95 \times 10^{-24}$ to $2.4 \times 10^{-24}$ s$^{-1}$ Pa$^{-3}$, corresponding to ice temperatures of $T = 0$ to -5 °C (Cuffey and Paterson, 2010, p.73). For the sliding parameter, we perform runs using $\beta = 4 \times 10^{-12}$ to 1 $\times 10^{-13}$ m a$^{-1}$Pa$^{-1}$.

The 'best-fit' combination is obtained by minimizing the Root Mean Square Error (RMSE) between the modelled ($H_{\mathrm{mod}}$)
and observed ($H_{\mathrm{obs}}$) ice thickness:

$$RMSE = \sqrt{\frac{\sum_{i=1}^{k}(H_{mod} - H_{obs})^2}{k}} \tag{9}$$

where $k$ is the number of vertices for which the RMSE is calculated.

Since the outlet glaciers Midtdalsbreen and Rembesdalskåka are of primary interest, we use the combined RMSE along their flowlines as the most important metric (Fig. 3). As an additional check, we also calculate the RMSE for ice thickness over the
entire ice cap (not shown here). Based on Fig. 3, we consider our 'best-fit' parameter combination to be $A = 2.0315 \times 10^{-24}$



s$^{-1}$ Pa$^{-3}$ ($T_{ice} = -1$°C ) and $\beta = 2 \times 10^{-12}$ m a$^{-1}$Pa$^{-1}$, though several parameter combinations produce similar RMSEs, as further discussed in Sect. 5.1.

### 3.2.3  Holocene

Lake sediment studies by Dahl and Nesje (1994) suggest that Hardangerjøkulen has been continuously present since c. 3800
BP, with some small local glacier activity during the millennium before. We therefore choose 4000 BP, with no ice cap present, as the starting point for our simulations.

Dahl and Nesje (1996) reconstructed summer temperature based on former pine-tree limits from southern Norway, as well as winter precipitation for the Hardangerjøkulen area based on lake sediment-derived ELAs and a well-established empirical relationship between winter precipitation and ELAs for Norwegian glaciers (Liestøl in Sissons, 1979; Sutherland, 1984). These reconstructions suggest a close to linear cooling and wetting trend from 4000 BP until the LIA, with some fluctuations superimposed (Velle et al., 2005a).

At 4000 BP, reconstructions (Dahl and Nesje, 1996; Velle et al., 2005b; Seppä et al., 2005) suggest temperatures 0.5 °C lower than present (favorable for glacier growth), and a drier climate (unfavorable for glacier growth). The combined effect implies a mass balance similar to present.

Based on this, we start from $\Delta B(t) = 0$ and thereafter linearly increase mass balance to 0.4 m w.e. over the period 4000 BP to 400 BP (1600 AD). The final value of $0.4$ m w.e. is chosen to produce an ice cap sized between the present-day and LIA extent. For this simulation, we use our 'best-fit' deformation and sliding parameters obtained from the calibration ensemble.

It is possible to refine or alternate this simple forcing in several ways. However, applying such changes based on poorly known past climatic and mass balance conditions adds additional uncertainty and unnecessary complexity.

### 3.2.4  Little Ice Age until present-day

Using our Holocene run ending at 1600 AD as initial conditions, we aim to reproduce the history of Hardangerjøkulen from the LIA until present-day, as well as to assess model sensitivity to choice of deformation and sliding parameters. For these purposes, we run the same parameter ensemble as used in the calibration process.

Since the mass balance record from Rembesdalskåka starts in 1963, mass balance has to be reconstructed for the period prior to this. A plausible mass balance history is therefore found from 1600 AD, through the LIA maximum in 1750 up to 1963, using a dynamic calibration (Oerlemans, 1997a, 2001). This approach is based on matching the model against the moraine evidence and length records of the outlet glaciers Midtdalsbreen and Rembesdalskåka, while adjusting $\Delta B(t)$ accordingly. As a starting point, the mass balance history obtained for Hardangerjøkulen by Giesen (2009) is tested. This history is then modified slightly to match the outlet glacier length records. However, we employ minimal tuning of the mass balance history, since a key aim is to investigate parameter sensitivity, and mass balance is arbitrary before 1963.

As a further constraint, modelled ice cap extents are compared with known past extents from maps and aerial photographs from the 1900s, and the modelled ice cap surface is compared with the 1995 DEM. The sliding parameter $\beta$ might change over time, since basal velocities may change with changes in surface melt, subglacial hydrology, basal roughness and other





factors. However, we keep our ensemble intact in both space in time, avoiding *ad-hoc* changes without physical foundation and validation, as explained in Sect. 3.1.1.

# 4 Results

## 4.1 Mid- to late Holocene evolution of Hardangerjøkulen

Starting from a present-day reference mass balance forcing at 4000 BP, we use our simple yet empirically based, linearly increasing mass balance (Fig. 6a) from 4000 BP (0 m w.e. anomaly relative to present-day) to 1600 AD (0.4 m w.e.). We demonstrate that the ice volume evolution for Hardangerjøkulen during the mid- to late Holocene was far from linear and differed between outlet glaciers (Fig. 6c). Starting from ice-free conditions, ice cap volume increases rapidly during the first ∼200 years (4000–3800 BP), then close to linearly between c. 3800–2300 BP (period A–B). Subsequently, starting at around 2300 BP, Hardangerjøkulen triples its volume over a period of 1000 years (B–C), before stabilizing at the end of the period (C–D).

Snapshots at times A–D reveal patterns of ice cap growth (Fig. 5). Initially, ice grows on high bedrock ridges above the ELA (Fig. 5a, also see Fig. 1). During the period of linearly increasing ice volume (A–B), Rembesdalskåka and Midtdalsbreen advance at similar rates. At this stage, Rembesdalskåka occupies an area with a gently sloping and partly overdeepened bed (Fig. 4).

After passing the lower edge of this overdeepening, Rembesdalskåka advances ∼3.5 km in 400 years (2300–1900 BP), corresponding to a length increase of 60 % (Fig. 4). In contrast, Midtdalsbreen is already at an advanced position in 2300 BP, and changes only modestly during this period.

Ice volume grows rapidly during 2300–1900 BP, however the advance and thickening of Rembesdalskåka can not alone explain this ice volume increase. Rather, the bulk of Hardangerjøkulen's volume increase during this period is due to ice cap growth in the east and southeast, where deep bedrock basins are filled with ice up to 400 m thick (Fig. 5d, see also Fig. 1).

We tested alternative mass balance forcings (faster rate of linear increase, and constant mass balance equal to the final value), and found the spatial pattern of ice cap growth robust for different forcings.

At the end of the spinup period (c. 1300–400 BP), outlet glaciers stabilize their frontal positions, and ice volume increase flattens out.

## 4.2 Hardangerjøkulen since the Little Ice Age

### 4.2.1 Parameter ensemble

The modelled state in 1600 AD is reached using our 'best-fit' parameter combination from the Holocene run. From 1600 AD, we run the model with our ensemble of sliding and deformation parameter combinations, for one specific mass balance history.

The ensemble modelled ice volumes at the LIA maximum (1750 AD) range from c. 12.7 to 17.4 km$^3$, and vary between 6.9 and 13.4 km$^3$ for the present-day (2008 AD; Fig. 6d).





### 4.2.2 Simulation using 'best-fit' parameters

Within the ensemble, the closest match with observed ice volume in 1961 and 1995 (black dots in Fig. 6d) is obtained by our independently calibrated 'best-fit' parameter combination. Modeled and observed ice volume in 1961 and 1995 differ by 0.10 and 0.22 km$^3$, respectively, or 1.1 and 2.3 % of total observed ice volume, respectively.

The LIA maximum ice volume using the 'best-fit' parameter combination is modelled to 14.8 km$^3$ (Fig. 6d). The simulation shows that Hardangerjøkulen has lost one-third of its volume between 1750 and present-day.

The simulated continuous ice volume history of Hardangerjøkulen, from 4000 BP through the LIA until today, including our ensemble from 1600 AD onwards, is shown in its entirety in Fig. 6cd.

The mass balance forcing used in Fig. 6 is derived by matching modelled and observed length variations of the outlet
glaciers Midtdalsbreen (NE) and Rembesdalskåka (SW). For the LIA maximum, it is not possible to obtain correspondence to observed lengths for both outlet glaciers simultaneously, not even by altering the dynamical parameters (Fig. 7). The mass balance history giving optimal results for Midtdalsbreen was chosen since its LIA maximum extent has been dated to 1750 AD, while no dates exist for Rembesdalskåka. In addition, bed topography is more accurately known for Midtdalsbreen than for Rembesdalskåka. When the model is calibrated against Midtdalsbreen's front variations, the LIA maximum length agrees
reasonably well with moraine evidence, whereas Rembesdalskåka is too short (Fig. 8).

Both modelled outlet glaciers are too short during the early 1900s, but the difference for Midtdalsbreen is only slightly larger than the model resolution (200 m). After 1960, the model-observation match is good, with differences being within the size of one mesh element (Figs. 7 and 8).

Consistent with the results for Midtdalsbreen and Rembesdalskåka, the lengths of the southwestern outlet glaciers at the
LIA maximum are underestimated in the model (Fig. 9a). The extent of the northeastern glaciers agrees well with moraine evidence.

While the outlet glaciers are too short around 1930 (Fig. 9b), the ice cap margin after 1960 is reproduced with a high degree of detail (Fig. 9cd). Most but not all discrepancies are close to the model resolution. One exception is the too small northwestern ice cap, however ice thickness in the missing area is small ($< 50$ m), so this mismatch contributes little in terms of total ice
volume. Modelled thickness in 1995 is generally in good agreement with the data, though the ice cap interior is somewhat too thin and the thickness along the eastern margin is overestimated (Fig. 9e).

## 5   Discussion

### 5.1   Sensitivity to sliding and deformation parameters

Running our parameter calibration ensemble, we aim to minimize the Root Mean Square Error (RMSE) between observed and
modelled present-day surface topography, yet several parameter combinations give similar RMSEs (Fig. 3). This is not surprising, since the parameters for ice deformation ($A$) and sliding ($\beta$) both depend on driving stress (Flowers et al., 2008; Zekollari et al., 2013). This highlights the challenge of picking a 'dynamically ideal' or even unique combination without empirical



knowledge about their relative importance, as noted by previous studies (Le Meur and Vincent, 2003; Adalgeirsdóttir et al., 2011; Zekollari et al., 2013). Given such ambiguities, and the fact that the impact of the deformation and sliding parameters may differ for varying mass balance regimes, we choose to keep our parameter ensemble intact from calibration to our historic runs, where we assess parameter effects on transient behavior from the LIA until today. Within this ensemble, we investigate

one 'best-fit' parameter combination (Table 1) in more detail.

Based on calibration (Fig. 3), model-observation misfit is sensitive to the choice of $A$ when using $\beta$ values corresponding to little sliding. Conversely, sensitivity to ice deformation is lower for faster sliding. With the lack of comprehensive observed velocities for validation, it is challenging to judge what range of the ensemble is more 'likely' or 'realistic', though we consider deformation parameters corresponding to temperatures of -5 °C less likely, since we expect this rheology to be too stiff for

a temperate ice cap. We therefore exclude simulations using this temperature from our historic ensemble. The magnitudes of modelled velocities for several parameter combinations are similar to the observed velocities available (Sect. 2.3.2), though there are too few measurements to constrain the parameters.

The ensemble spread for ice volume in the historic run from LIA until today is large (Fig. 6). However, 22% of the ensemble spread for present-day (year 2008) can be attributed to an ice rheology corresponding to -3°C, which may not be soft enough

for Hardangerjøkulen's temperate ice.

During the years subsequent to 1600 AD, after the change of dynamical parameters, the ice cap response is a combined effect of climate forcing and adjustment to new parameter values. However, the period 1600–1710 AD can be viewed as an additional short spinup phase for the historic simulation, since we keep the mass balance constant at the end value of the Holocene simulation ($\Delta B(t) = 0.4$ m w.e.) during this period.

For the historic run, we observe that the ensemble spread in surface elevation is larger in the vicinity of the ELA than at the periphery (Fig. 7). This phenomenon can be explained by the fact that a change in $\beta$ or $T_{ice}$ leads to either an increase in ice velocity (if $\beta$ and/or $T_{ice}$ increases) or a decrease in ice velocity (if $\beta$ and/or $T_{ice}$ decreases). When the velocity increases, it takes a shorter amount of time for the ice to flow from the summit to the ELA, and therefore the ice thickness at the ELA is smaller, since it has not spent as much time in the accumulation zone. On the other hand, ice will also flow faster downstream

and will therefore spend less time from the ELA to the terminus for the same ablation rate. Similarly, for slower velocities, we expect that ice is thicker at the ELA, but the deviation in ice thickness decreases as we reach the glacier terminus.

In agreement, the ensemble spread for the front position itself is small. The latter is also due to a combination of a highly negative mass balance at this elevation, and ice not flowing fast enough (in the order of 50 m a$^{-1}$ or less) to replace the mass lost, preventing the front from moving to the next mesh node (which lies 200–300 m ahead).

A future expansion of this work, outside the scope of this study, would be a multiple regression of the dynamical parameters for Hardangerjøkulen and its outlet glaciers. This could disentangle whether their importance changes over time, for example depending on mass balance regime or whether the glacier is retreating or advancing.

By imposing changes in dynamical parameters and exploring their model sensitivities, we can estimate how dynamical changes affect ice masses. For example, ice may go from cold- to warm-based or vice versa, or transient changes in basal

conditions may change sliding speeds. Given recent advances in data assimilation, including methods to estimate basal slip-





periness for present-day ice sheets (e.g. Morlighem et al., 2010), it would be interesting to see how stable the obtained friction maps are through time. Such issues should be of concern for model studies of future as well as past ice sheet behavior.

## 5.2 Sliding in previous studies

Many previous studies (e.g. Payne, 1995; Ritz et al., 1996; Payne, 1999; Payne et al., 2000; Flowers et al., 2008; Le Brocq
et al., 2009; Giesen and Oerlemans, 2010; Adalgeirsdóttir et al., 2011; Clason et al., 2014) employing the SIA have used a spatially and temporally fixed sliding parameter in a Weertman-type sliding law (Weertman, 1957).

Le Brocq et al. (2009) found that the sliding parameter $\beta$ for West Antarctica varied over five orders of magnitude in response to available water, ranging from $1 \times 10^{-5}$ to $1 \times 10^{-1}$ m a$^{-1}$Pa$^{-1}$. In our study, the 'best-fit' $\beta$ in the ensemble is $6.3 \times 10^{-5}$ m a$^{-1}$Pa$^{-1}$, while the ensemble ranges from $3.16 \times 10^{-4}$ to $1.26 \times 10^{-5}$ m a$^{-1}$Pa$^{-1}$. The sliding parameter we use
for Hardangerjøkulen is thus in the lower range of what Le Brocq et al. (2009) suggested, corresponding to areas away from ice streams, which is the type of environment we would expect most similar to Hardangerjøkulen. Compared to the values Payne (1995) and Payne (1999) used to model ice sheets, the Hardangerjøkulen ensemble values are 1-2 orders of magnitude lower (i.e. less slippery). However their deformation parameters $A$ are 1-2 orders of magnitudes higher than ours (i.e. softer), and since both the sliding and deformation relation used are linear with respect to velocities, the combined effect is similar.

Flowers et al. (2008) simulated Holocene behavior of the Langjökull ice cap on Iceland using $\beta = 2.5 \times 10^{-4}$ m a$^{-1}$Pa$^{-1}$, which is within our ensemble range. Somewhat in contrast to this study, they noted a low sensitivity to $\beta$. They attributed this insensitivity to the lack of a seasonally driven surface velocity cycle. Conversely, seasonal speed-ups have been observed at Hardangerjøkulen (Willis, 1995; Willis et al., 2012). It is therefore not surprising that Hardangerjøkueln is more sensitive than Langjökull to the choice of sliding parameter.

In contrast to this study, some previous studies of smaller ice masses have used a non-linear Weertman-type sliding (Le Meur and Vincent, 2003; Jouvet et al., 2011; Adalgeirsdóttir et al., 2011; Zekollari et al., 2013). While a direct comparison of our sliding parameter $\beta$ is not possible, they note that several combinations of their sliding and deformation parameter give similar results, in line what we find here, as discussed in Sect. 5.1.

Hubbard et al. (2006) used field evidence to constrain model experiments for the Last Glacial Maximum in Iceland, and
obtained a shallow, dynamic ice sheet, associated with high sliding. Using similar methods for the Younger Dryas ice sheet in Scotland, Golledge et al. (2008) noted subtle but consistent patterns when varying sliding values. Specifically, their modelled ice sheet became thinner but more extensive with increased sliding, consistent with Hubbard et al. (2006)'s and our findings. Thus, for whatever the cause, high sliding seems to be associated with, perhaps sometimes a prerequisite for, a shallow geometry.

It is possible to adjust the basal sliding parameter over time. However, in the *ad-hoc* formulation used, factors like surface meltwater supply, thermal regime, bed roughness, and the type of drainage system are lumped together in the parametrization and may change differently over time. We therefore consider transient adaptation of basal slipperiness in this study speculative rather than insightful. Alternative approaches to our sliding formulation include a non-linear Weertman sliding law (e.g. Pattyn,



2002; Le Meur and Vincent, 2003; Jouvet et al., 2011), effective pressure-dependent sliding (e.g. Schoof, 2005; Tsai et al., 2015), or relating $\beta$ to surface melt or another climate variable (Greve and Otsu, 2007; Clason et al., 2014).

Better knowledge of the bed properties at Hardangerjøkulen by means of radar, seismics or borehole studies, along with modeling of the subglacial drainage system, would be steps toward understanding the (transient) behavior of basal slipperiness.

## 5.3 Uncertainties in mass balance

For the LIA maximum, the terminal moraine at Midtdalsbreen is dated to 1750 AD, while the moraine at Rembesdalskåka is not dated, but assumed to be formed at the same time. The true maximum for Rembesdalskåka may however have had a different timing, though the model resolution is probably too coarse to investigate details of such asynchronous advances, let alone the uncertainty in the climatic forcing.

The challenge to accurately model Rembesdalskåka and Midtdalsbreen simultaneously during the LIA (Fig. 7) may be related to the mass balance formulation used. As implemented here, mass balance is only a function of elevation. Rembesdalskåka is facing the prevailing westerly wind direction and is expected to receive more snow than Midtdalsbreen. To account for a west-east gradient in winter accumulation, a potential improvement would be to let mass balance vary spatially.

Unfortunately, the two single years (2001-02; Krantz, 2002) with mass balance measurements on Midtdalsbreen are not enough to systematically assess differences in the mass balance regimes of Rembesdalskåka and Midtdalsbreen, though differing mass balance regimes are suggested by Andreassen and Elvehøy (2001), who calculated surface elevation change from 1961 to 1995. West-east mass balance gradients have also been proposed as an explanation for differing glacier reconstructions between the southwestern margin (Nesje et al., 1994) and the northeastern margin of the ice cap (Dahl and Nesje, 1994). Further glacier reconstructions based on multiproxy approaches on lacustrine sediments (e.g. Vasskog et al., 2012) could give more insight into differing continentality of the outlet glaciers of Hardangerjøkulen.

A horizontal precipitation gradient at Hardangerjøkulen was assumed by Giesen and Oerlemans (2010). This effect was added artificially based on meteorological station data on respective sides of the ice cap rather than from *in situ* measurements of mass balance. Quantification of the spatial variability of accumulation through further snow and mass balance studies would be valuable to better understand the climatic response of Hardangerjøkulen.

Besides imposing horizontal mass balance gradients, the mass balance maximum in the vertical profile (~1775 m a.s.l.) can been adapted in time as ice cap geometry changes. However, Giesen (2009) showed that temporally shifting the mass balance maximum according to summit elevation at Hardangerjøkulen plays a minor role when mass balance is slightly positive, but gives a more sensitive ice cap for negative mass balances. Our approach with a static mass balance vertical profile should therefore be regarded as conservative when it comes to mass balance sensitivity. Moreover, we found that including a mass balance-altitude feedback was a crucial feature of the mass balance formulation (Sect. 5.8), in agreement with Giesen (2009).

It is not clear why observed mass balance decreases at the uppermost elevations (Fig. 2), but a likely explanation is snow redistribution by wind. Effects of snow erosion and redeposition may be parametrized based on surface curvature, which is a good indicator of regions with wind-induced snow redistribution (Blöschl et al., 1991; Huss et al., 2008). Giesen (2009) tested





a surface-curvature approach for Hardangerjøkulen, however the plateau was too flat for snow redistribution to occur in the model.

Glaciological measurements of mass balance have inherent uncertainties and biases, related to instrumentation, survey practices and techniques (Cogley et al., 2011). Andreassen et al. (2015) performed a reanalysis of glaciological and geodetic mass balance for Norwegian glaciers, including Rembesdalskåka. For the period 1995-2010, they found a more negative geodetic mass balance (-0.45 m w.e.) than the glaciological one used in this study. An additional simulation with this more negative mass balance for the final years of our simulation (1995–2008) shows that the effect on ice volume is c. 0.5 km$^3$, or 5.3 % of modelled ice volume in year 2008.

Though the simple mass balance formulation used in this study works well, there are uncertainties associated with applying present-day observed mass balance profiles in different climates. It is reasonable for smaller ice cap changes, but may be less accurate when Hardjangerjøkulen grows or shrinks considerably. Surface topography changes may affect accumulation patterns, especially if the ice cap splits into individual outlet glaciers, something not captured in our implementation.

In a warmer climate, like that of the mid-Holocene, the melt season will also be longer. This implies that surface albedo will be lower for a longer time period every year, because an earlier melt season onset will expose bare ice earlier in the summer (Oerlemans and Hoogendoorn, 1989). This is a positive feedback, since lower albedo means that more melt occurs at a given temperature. Since our mass balance forcing is derived from present-day climate, our simulations may underestimate the climate sensitivity during the warm mid-Holocene and the predicted warm future.

Finally, solar insolation patterns may also change with strongly altered ice cap geometry, for example by shading effects of valley walls. Nevertheless, Giesen and Oerlemans (2010) accounted for longer melt seasons and solar insolation changes when applying a spatially distributed energy balance model until 2100 AD. They did not find very large changes in the mass balance gradient, indicating that the transferability of today's mass balance profile is robust on these time scales. Furthermore, solar irradiance at 4000 BP, when we start our simulation, was at most 5% larger in the summer months than today (Giesen, 2009), and will thus not have considerable effect on mass balance.

## 5.4 Impact of ice dynamics

This study supports previous glacier modeling exercises (e.g. Le Meur et al., 2007; Guðmundsson et al., 2009; Giesen and Oerlemans, 2010; Adalgeirsdóttir et al., 2011), as well as theoretical comparisons between SIA and Full Stokes (FS) models (Leysinger-Vieli and Gudmundsson, 2004; Hindmarsh, 2004; Gudmundsson, 2008), in that SIA is viable to use if interests are climatic rather than ice dynamics.

SIA inaccuracy is a candidate for explaining model-observation differences in regions with steeper bedrock slopes at Hardangerjøkulen, conditions where SIA has been shown to be inaccurate in idealized studies (Le Meur et al., 2004). Work by Hindmarsh (2004) and Gudmundsson (2008) using idealized glacier geometries showed that the SIA accurately represents large scale flow, in the absence of significant basal sliding. By investigating a small valley glacier in the Canadian Rocky Mountains and neglecting basal sliding, Adhikari and Marshall (2013) suggested that SIA performs well in less 'dynamic' settings, while the results compared to HO/FS diverge for more 'dynamic' situations.





It is challenging to assess how much sliding there could be before SIA validity deteriorates, but it likely depends on the climatic and glaciological setting. Moreover, bed topography data used in many studies are uncertain. Care should therefore be taken before drawing too many conclusions on SIA accuracy based on the bedrock slopes in different areas of Hardangerjøkulen.

Given our interest in the climatic response, and the lack of fast-flowing areas on Hardangerjøkulen, we consider SIA to be a valid choice for this study. Nevertheless, since we have not compared the SIA with HO/FS for Hardangerjøkulen, we cannot conclude how accurate SIA is for Hardangerjøkulen in particular and small ice caps in general. To understand the significance of higher-order ice mechanics, further similar studies are needed for different dynamic, climatic and topographic settings.

Finally, because of the relatively poor process knowledge on processes at glacier beds, uncertainties regarding spatial and
temporal patterns of basal sliding may be larger than the difference between a simple (SIA) and a physically more complete (HO/FS) ice flow model. To understand glacier behavior under climate change, we advise that ice flow model intercomparisons of *real* glaciers and ice caps should consider the sensitivity to both ice deformation and basal sliding (and of course, mass balance).

### 5.5   Implications of modelled Holocene evolution

In the early part of the modelled period (c. 4000 - 3800 BP), ice grows preferentially on high bed topography, and earlier on Midtdalsbreen/Blåisen than in the present-day basin of Rembesdalskåka (Fig. 5, also see Fig. 1). While the model resolution here is coarse (300-500 m), we expect that ice dynamics at this stage plays a minor role, since the ice present is divided over several small (< 2 km long, < 100 m thick) glaciers. Instead, the initial ice growth at high bed ridges is due to build-up of ice above the present-day ELA, which is used as initial mass balance forcing at 4000 BP.

Reconstructions around southern Norway show that glaciers did not survive the mid-Holocene thermal maximum (e.g. Bakke et al., 2005; Nesje, 2009). In agreement, pollen-based reconstructions from western Norway suggest a drop in summer temperatures at 4000 BP (Bjune et al., 2005).

We are aware of the limitations of the SIA in the steep terrain where Rembesdalskåka terminates during the period of fast ice volume increase (c. 3800–2300 BP, Fig. 6c). Therefore, the actual rate of advance may differ from what is modelled here.
However, we expect that this section of Rembesdalskåka's bed fosters a more dynamic glacier than the overdeepened central part, so the simulated fast advance in this area is not surprising.

During the period of modelled rapid ice cap growth (c. 2300–1300 BP), reconstructed precipitation in western Norway is slightly lower than the general increasing trend applied here (Dahl and Nesje, 1996; Bjune et al., 2005), coincident with glacier reconstructions from southern Hardangerjøkulen indicating a slight decrease in glacier size (Nesje et al., 1994). In our mass
balance forcing, we deliberately smoothed out any variability around the general trend, since our aim is to understand the first order aspects of ice cap growth through time. Moreover, there is to our knowledge no geomorphological or other evidence that can be used as tie points for ice cap extent or volume during this period. Imposing short variations in mass balance would therefore add further uncertainty rather than improve our understanding of the behavior of Hardangerjøkulen and the first order impact of bed topography.



### 5.6 Non-linearity, asymmetry and paleoclimatic relevance

Comparing the ice volume evolution for three of the outlet glaciers (Rembesdalskåka, Midtdalsbreen, Blåisen), we find that they do not grow in the same fashion (Fig. 6c). Midtdalsbreen's ice volume increases linearly over time, while Rembesdalskåka and Blåisen have distinct jumps in ice volume, related to their bed topography. The importance of bedrock troughs and overdeepenings is further illustrated by Hardangerjøkulen's non-linear volume increase c. 2300–1300 BP, a period when volume increases faster than area (Fig. 10). In other words, ice during this period is thickening rather than expanding horizontally. This can largely be explained by ice growth in subglacial valleys in the eastern and southeastern parts of the ice cap (Fig. 1). These bed depressions fill up quickly because ice flow converges into them from surrounding high bedrock ridges, and the mass balance-altitude feedback amplifies the ice thickening.

The initial present-day mass balance forcing ($\Delta B(t) = 0$ m w.e.) at 4000 BP likely explains the rapid increase in ice volume over the first few hundred years, since this forcing essentially represents a step change in mass balance at 4000 BP. However, this effect diminishes after a few hundred years, after which the response is due to the linear mass balance forcing. $\Delta B(t) = 0$ m w.e. starting from ice-free conditions produces a steady-state ice volume of only $\sim 2$ km$^3$ (Fig. 11), a volume which is exceeded at 3300 BP, so any additional ice volume cannot be explained by the initial step change in mass balance at 4000 BP. Most importantly, the non-linear ice volume response between 2300–1300 BP is thus entirely forced by the linear mass balance increase during this period.

Analogous to the Holocene simulations, we also performed experiments with a slowly *decreasing* mass balance over multiple millennia (from $\Delta B(t) = 0.4$ to 0 m w.e.), allowing the ice cap to dynamically adjust, starting with the 1600 AD ice cap state. We find that the western ice cap disappears first, while ice in the eastern part of the ice cap is more persistent (not shown here).

It is striking that the western and northern parts of the ice cap grow first and disappear first, whereas the eastern part grows last and disappears last. This asymmetry illustrates that proxy records representing different parts of an ice cap may lead to substantially different conclusions about ice cap size through time.

Previous work has highlighted glacier hypsometry, overdeepenings and proglacial lakes in altering glacier *retreat* to climate forcing (Kuhn et al., 1985; Jiskoot et al., 2009; Adalgeirsdóttir et al., 2011). Adhikari and Marshall (2013) and Hannesdóttir et al. (2015) showed that overdeepened basins loose mass by thinning rather than retreat. Here we suggest that a similar behavior applies to an *advancing* glacier. In particular, overdeepened areas delay frontal advance and lead to preferential glacier thickening. However, note that the effect of higher order stresses, not captured by our simplified dynamic model, may be more important for an advancing glacier (Adhikari and Marshall, 2013).

Notably, our experiments show that a gradual (linear) climatic change results in a non-linear change in ice volume. Given that we would like to understand past climates and perform future predictions, our results clearly pinpoint that we should assess underlying mechanisms and resulting feedbacks, rather than extrapolate a climatic forcing and glacier change concurrently through time.

In general, steeper outlet glaciers with a high ablation rate adjust more rapidly to climate than thick, gently sloping ice masses with a lower melt close to the terminus (Johannesson et al., 1989; Harrison et al., 2001). Elsberg et al. (2001) and




Harrison et al. (2001) discuss time scales of response in light of different glacier characteristics. Consistent with our findings, they suggest that the effect of surface elevation is critical for the response of gently sloping ice masses.

Based on theoretical considerations accounting explicitly for mass balance changes, and implicitly for ice dynamics, they find a useful ratio for glacier response time to be $\dot{G}_e \frac{H}{\dot{b}_e}$, where $\dot{G}_e$ is the (linear) mass balance rate of change with altitude, $H$ is an characteristic thickness scale and $\dot{b}_e$ an 'effective' balance rate close to the terminus. They suggest that if this ratio approaches unity, the sensitivity as they have defined it becomes high, and errors can become large in the calculated response, because surface elevation and area feedbacks nearly cancel each other out.

For Hardangerjøkulen, mass balance does not vary linearly with elevation for the entire elevation range, but typical values for the 1963-2007 period for the linear part of the gradient we use are $\dot{b}_e$ = -6.5 m ice eq. a$^{-1}$ and $\dot{G}_e$ = 0.0097 a$^{-1}$. A characteristic thickness scale for Hardangerjøkulen is 150–200 m. Using the formula from Harrison et al. (2001), this gives a ratio of around 0.2 to 0.3. For a LIA situation, the characteristic thickness may have been 250 m, with a weaker ablation rate at the terminus, giving ratios of 0.5 to 0.6, assuming the same vertical mass balance gradient. These values are all well below Harrison et al. (2001)'s problematic ratio close to unity. However, it should be noted that their theory assumes that area can react instantaneously to volume, thus remaining in-phase, something we do not observe for several outlet glaciers and periods of Hardangerjøkulen's late-Holocene history. Giesen (2009) attempted to calculate response times for Hardangerjøkulen and its outlet glaciers, but could not define a characteristic response time because of the high sensitivity and variation between outlet glaciers.

Our simulated preferential ice cap growth on the northern and western side, illustrated in Fig. 5b at 2300 BP, is in line with reconstructions showing an early glacierization of the north (Dahl and Nesje, 1994) versus the south (Nesje et al., 1994), though there is potential for further studies investigating the spatial asymmetry in more detail.

Importantly, glacier reconstructions using proglacial lake sediments are generally based on assumed changes in glacier (erosive) area rather than volume (Hallet et al., 1996), while we show that volume and area can become decoupled for several hundred years at a time (Fig. 10), for example when the largest outlet glacier Rembesdalskåka was situated on overdeepened parts of its bed. We also demonstrate that the degree of volume-area coupling varies for different outlet glaciers, implying that each outlet glacier should be considered individually. For example, a differing response to identical climate forcing is illustrated when Midtdalsbreen advances only modestly from 2300–1300 BP (Fig. 5b-d), while Hardangerjøkulen triples its ice volume during the same period due to ice growth occurs elsewhere (mainly in the east, south and southwest).

The wider implication of our results is that glaciers have different climate sensitivities depending on where the ice margin is located and what is underneath it. It follows that increased proglacial lake sediment input may not indicate an advancing glacier, but merely a thickening and increasingly erosive glacier occupying an overdeepened part of its bed. Similarly, assuming that changes in sedimentary input reflect area change rather than volume, and extracting climatic signals from a preferentially thinning glacier with a stagnant front, may be challenging at best.

Our study proposes a reassessment of some glacier reconstruction methodologies, in particular those using sediments from proglacial lakes. We advise that such studies should infer past climatic and glacier states not exclusively using a linear assump-





tion between sediment input and glacier basin size. Ideally such records would be accompanied by (modelled or empirical) knowledge of the interaction between past ice dynamics, sedimentation, mass balance and geometry.

## 5.7 LIA maximum and initial conditions

Our simulations show that Hardangerjøkulen has lost one-third of its volume since 1750 AD. The modelled LIA maximum volume of ∼14.8 km$^3$ (Fig. 6) is challenging to validate, since we do not know the surface topography at this time.

We are aware that bed topography at Hardangerjøkulen is uncertain in places, though less so for Midtdalsbreen and Rembesdalskåka, which are of prime interest. Moreover, the proglacial lake in front of Rembesdalskåka may have modulated LIA frontal behavior, as suggested for Icelandic glaciers (Hannesdóttir et al., 2015).

The exact history of Hardangerjøkulen as a whole will thus unavoidably differ from our model results. However, given the limited knowledge about ice cap activity between the ice-free conditions at 4000 BP and the LIA maximum around 1750 AD, we consider our continuous model reconstruction to be a good first estimate of how Hardangerjøkulen grew from nothing to its most extensive state during the LIA.

Moreover, we have provided a plausible ice cap history over several thousand years as the starting point for our simulations from the LIA until today. This is in our view a step forward from several previous studies (e.g. Giesen and Oerlemans, 2010; Adalgeirsdóttir et al., 2011; Zekollari et al., 2014), that reach desired initial LIA conditions by perturbing a present-day ice cap.

## 5.8 Mass balance sensitivity

To investigate the sensitivity of present-day Hardangerjøkulen to 'future' changes in mass balance, steady-state experiments were performed with present-day ice cap topography as the starting point. The ice flow model is first forced without a mass balance anomaly until steady-state, using our 'best-fit' parameters from the calibration (Sect. 3.2.2). From this state, we perturb the mass balance by anomalies between -0.5 and +0.5 m w.e., and run the model to a new equilibrium.

These experiments show a close to linear relationship between mass balance perturbation and ice volume response (Fig. 11), until the point when the mass-balance feedback becomes too strong and the ice cap disappears entirely for more negative anomalies.

Our simulations show that Hardangerjøkulen is highly sensitive to climate warming (Figs. 11 and 12). In particular, the ice cap is bound to disappear almost entirely for a mass balance anomaly of -0.3 ± 0.2 m w.e., relative to the mean mass balance over the period 1963-2007. These results are consistent with those of Giesen (2009), who used a SIA model (Van Den Berg et al., 2008) with different implementation of basal sliding and ice deformation and without local mesh refinement. Similar experiments with Nigardsbreen, southwestern Norway (Oerlemans, 1997a), and Franz Josef Glacier, southwestern New Zealand (Oerlemans, 1997b), both located in maritime climates, show much smaller volume and length responses, illustrating the extreme mass balance sensitivity of Hardangerjøkulen.

To investigate the role of the mass balance-altitude feedback in the ice cap response, we performed additional experiments excluding this feedback by keeping the mass balance field fixed at the present-day surface topography. Using this setup, a close





to linear relationship between mass balance changes and steady-state ice volume was found, and the ice cap was less sensitive to mass balance changes. For example, without the feedback, half of present-day ice volume (4.9 km$^3$) is still present for a mass balance anomaly of -0.5 m w.e., relative to 1963–2007 (not shown here). In contrast, when including the feedback, the ice cap disappears entirely for this mass balance anomaly.

We can view our results in light of future climate change. Compared to the period 1963-2000, with a mass balance close to zero, the mean mass balance in the last decade has decreased to -0.3 m w.e. Since Hardangerjøkulen was in approximate balance over the past decades, this decrease primarily reflects changes in meteorological conditions, and not dynamical adjustments. This mass balance change is small compared to the interannual variability in the net mass balance. Even if the mass balance does not become more negative in the future, Hardangerjøkulen is bound to disappear, although it will take 750 years (Fig. 12).

As evident from Collins et al. (2013), we expect a warming scenario for the future. Giesen and Oerlemans (2010) used an energy balance model to simulate the mass balance of Hardangerjøkulen for the next 100 years, using climate scenarios for southern Norway. Their modelled future mass loss largely exceeds that observed for the last decade. For example, for a 'realistic' scenario with a temperature rise of 3°C and 10 % rise in precipitation relative to the normal period 1961-1990, the net mass balance of Hardangerjøkulen as a whole was estimated to be -4.10 m w.e. in 2086, i.e. a mass balance more than 10 times

more negative than observed for the last decade. They also coupled their mass balance model to a SIA model (Van Den Berg et al., 2008), and suggested that Hardangerjøkulen will vanish almost completely before 2100. The rate of mass balance change in the 21st century is so large that the role of the mass balance-altitude feedback and ice dynamics in the modelled ice volume change is relatively small.

Similar conclusions have been reached for glaciers in Iceland (Adalgeirsdóttir et al., 2006; Guðmundsson et al., 2009; Adal-

geirsdóttir et al., 2011), French Alps (Le Meur et al., 2007), Swiss Alps (Jouvet et al., 2011) and Canadian Rocky Mountains (Clarke et al., 2015).

The high sensitivity to mass balance found for Hardangerjøkulen supports changes inferred for the Holocene. Abrupt changes are evident from lake sediment records of glacial activity both at the northern (Dahl and Nesje, 1994) and the southern (Nesje et al., 1994) sides of the ice cap. One example is the so called *Finse event*, when an advance to a maximum beyond that of

present-day of the northern Blåisen outlet glacier ∼8300 BP was followed by a complete melt-away of this glacier within less than a century. Since future warming is projected to be much larger than the changes during this period, a complete disappearance of Hardangerjøkulen is likely. Further studies on similar ice masses, integrating proxy data and modeling efforts, are needed to shed light on the relevant processes involved in such abrupt changes of ice caps and glaciers to climate change.

## 6   Conclusions

We have used a two-dimensional ice flow model with mesh refinement to simulate the evolution of Hardangerjøkulen ice cap since the mid-Holocene, from ice-free conditions up to the present-day. Until the LIA, the model is forced by a mass balance based on reconstructions of temperature and precipitation. From the LIA onwards, an optimized mass balance history is employed, and direct mass balance measurements are used after 1963.



We used the Shallow Ice Approximation (SIA) for ice flow and an ensemble approach to assess sensitivity to sliding and ice deformation parameters during both calibration and transient runs.

We show that the effect of the sliding parameter depends on what deformation parameter is used; the softer the ice, the more important is the sliding parameter. Moreover, we find it challenging to pick a unique, 'dynamically ideal' parameter combination. We therefore suggest that parameter ensembles used to calibrate models may as well be kept for transient simulations in the way presented here, if computationally feasible.

Our simulations show that Hardangerjøkulen evolved from no ice in the mid-Holocene to its LIA maximum in different stages, where the fastest stage (2200–1200 BP) involved a tripling of ice volume over only 1000 years.

Notably, our linear climate forcing during this time gives a non-linear response in ice cap volume and area. This growth occurs in a spatially asymmetric fashion, where Midtdalsbreen reaches its maximum first, while advances of Rembesdalskåka and the eastern ice cap are delayed. In contrast, an opposite spatial asymmetry is found for a disappearing ice cap. This response is linked to local bed topography; in particular, we highlight that the presence of an overdeepening delays glacier advance. We also illustrate that the degree of volume-area coupling for outlet glaciers as well as the whole ice cap varies both temporally and spatially.

Our simulations thus provide new insight relevant for paleoglaciological and -climatic studies assessing the influence of past ice dynamics and geometry. These considerations are also important for future predictions, though ice dynamics may become less important if the rate of future mass balance change is large.

Instead of perturbing the present-day ice cap as in previous studies, we reach initial conditions for our historical simulations starting at the LIA by modeling the Holocene ice cap history. Following the simulated Holocene growth of Hardangerjøkulen, we successfully reproduce the main features of the LIA extent of the main outlet glaciers, given temporal and spatial uncertainties in moraine evidence. In the early 1900s the simulated glacier positions are slightly underestimated, whereas the ice extent closely resembles the observed margins available starting from 1960, and the surface topography fits well with the 1995 surface survey.

Hardangerjøkulen is found to be highly sensitive to mass balance changes, consistent with previous studies of both the past and the present. A shift by only -0.3 ±0.2 m w.e. relative to the 1963–2007 reference mass balance induces a strong mass balance-altitude feedback and completely melts away the ice cap.

Several factors may affect the validity of the model for situations which differ largely from the present-day situation, including mass balance distribution for a greatly altered geometry, ice albedo feedbacks and glacio-hydrological changes. More work is needed to better constrain the time scale and relative importance of these aspects.

Although our simple mass balance implementation may be refined and physical complexity may be added to our sliding formulation or ice dynamical approximation, we consider our findings robust on the climatic time scales studied here.

*Acknowledgements.* We wish to acknowledge NVE for access to mass balance data, glacier outlines, and surface and bed DEMs. Thanks also to Norwegian Meteorological Institute for providing climate data (eklima.no). We are also grateful to Atle Nesje for sharing detailed knowledge about Hardangerjøkulen, and for reading the manuscript prior to submission. The research leading to these results has received





funding from the European Research Council under the European Community's Seventh Framework Programme (FP7/2007-2013) / ERC grant agreement 610055 as part of the ice2ice project. H.Å. was supported by the Research Council of Norway (project no. 229788/E10), as part of the research project Eurasian Ice Sheet and Climate Interactions (EISCLIM), and has also received travel support from BKK AS.





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



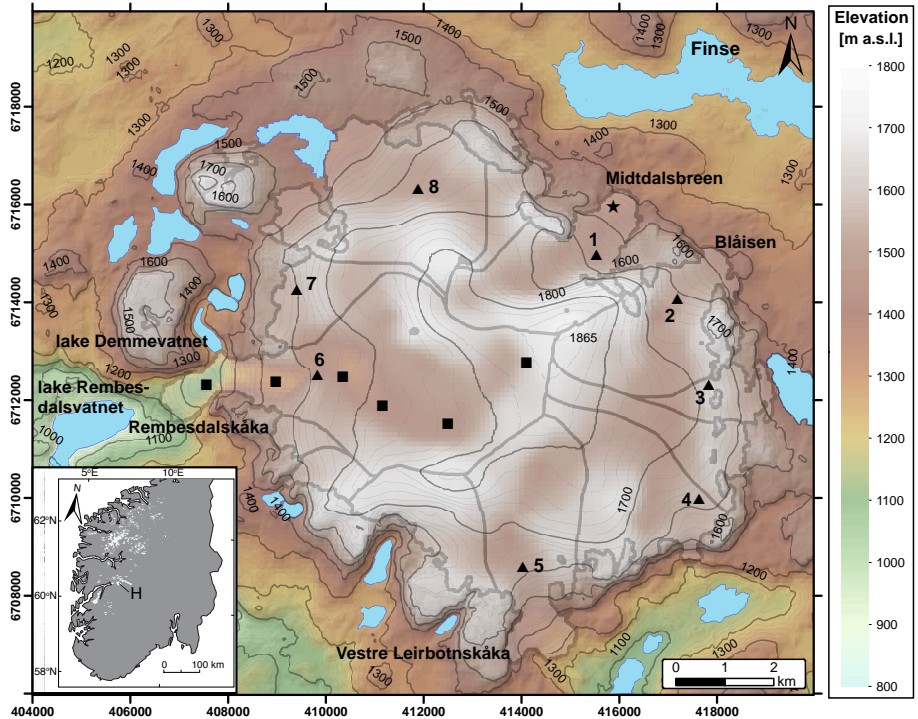

**Figure 1.** Bed (coloring) and surface (contours) topography of Hardangerjøkulen ice cap. Contour interval is 20 m and created from a digital elevation model by Statens Kartverk, 1995. The reference system is UTM zone 32N (EUREF89). Ice cap outline and drainage basins from 2003 are indicated (data from Cryoclim.net), as well as surrounding lakes (drawn after Statens Kartverk N50 1:50 000). Shown are GPS positions for velocity measurements (numbered triangles), mass balance stakes from NVE (squares) and location of the automatic weather station (star). Inset: map of southern Norway showing the location of Hardangerjøkulen (H).

**Table 1.** Constants and parameter values used in this study.

| Parameter | Symbol | Unit | Value |
|---|---|---|---|
| Ice density | $\rho_i$ | $\mathrm{kg\,m^{-3}}$ | 917 |
| Gravitational acceleration | $g$ | $\mathrm{m\,s^{-2}}$ | 9.81 |
| Flow factor | $A$ | $\mathrm{s^{-1}\,Pa^{-3}}$ | $0.95 \times 10^{-24}$ to $2.4 \times 10^{-24}$ |
| Sliding parameter | $\beta$ | $\mathrm{m\,s^{-1}\,Pa^{-1}}$ | $4 \times 10^{-12}$ to $1 \times 10^{-13}$ |
| Sliding law exponent | $m$ | | 1 |
| Glen's law exponent | $n$ | | 3 |
| Mesh resolution | $\Delta x$ | m | 200-500 |
| Time step | $\Delta t$ | a | 0.02 |





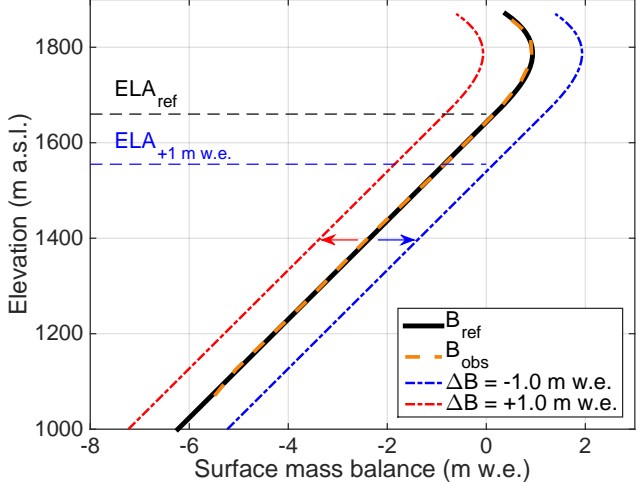

**Figure 2.** Reference net surface mass balance ($B_{ref}$) profile used in the model runs, based on the mean observed ($B_{obs}$) profile for 35 of the 45 years 1963-2007. At lower elevations, a linear gradient is used; for the highest elevations, a third-order polynomial is fitted to the observed values. Shown are also $\Delta B(t)$ = -1.0 and +1.0 m w.e., examples of how temporal mass balance changes are imposed (Eq. 8), along with corresponding ELA's. For -1.0 m w.e., mass balance is negative at all elevations, thus ELA is above the summit. Data from NVE.

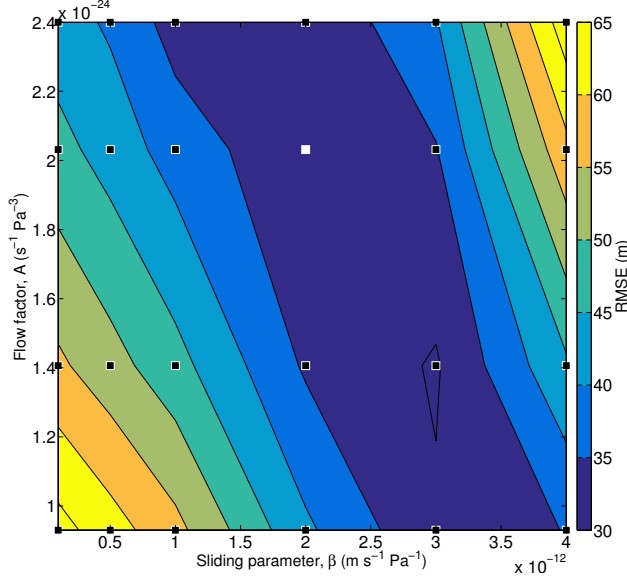

**Figure 3.** Root mean square error (RMSE) between modelled and observed present-day ice thickness along the flowlines of Midtdalsbreen and Rembesdalskåka, using an ensemble of sliding ($\beta$) and rheology ($A$) parameters. Shown are parameter combinations (black squares) and the 'best-fit' parameter combination used in subsequent runs (white square).



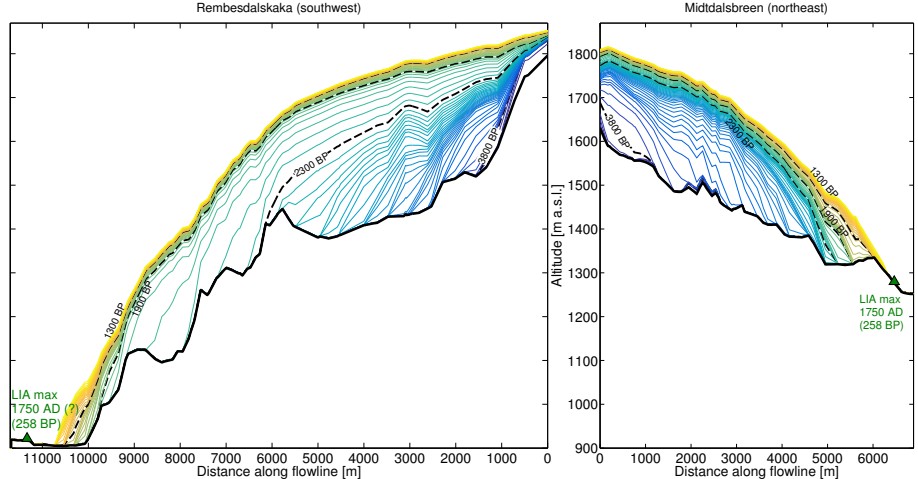

**Figure 4.** Modelled surfaces from 4000 BP to 1600 AD, starting with no ice cap, shown every 50 years from older (dark blue) to younger (yellow). BP ages are relative to 2008 AD. Note that the top of Rembesdalskåka (Hardangerjøkulen's summit) does not coincide with the top of Midtdalsbreen's flowline (see Fig. 9d).

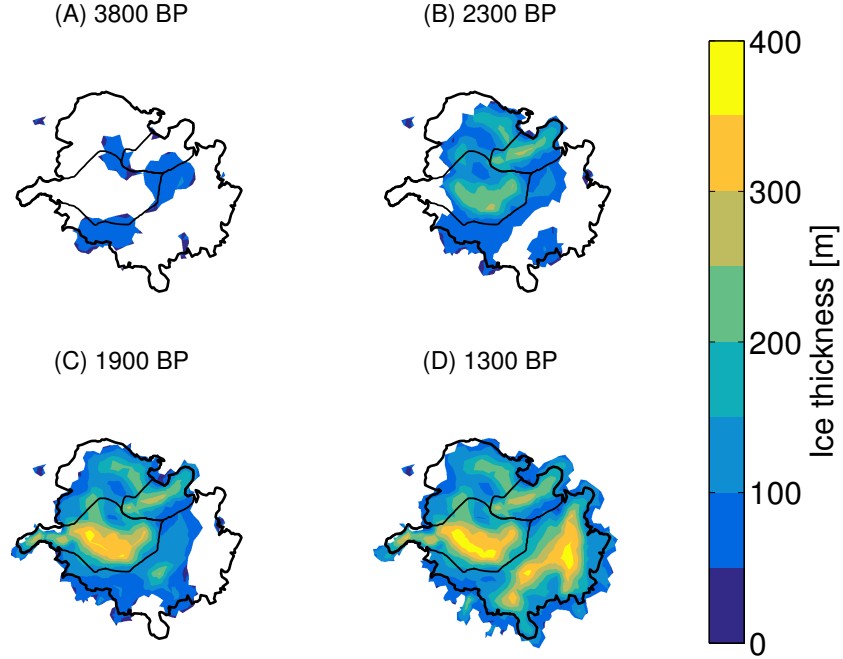

**Figure 5.** Modelled ice thickness at (A) 3800, (B) 2300, (C) 1900 and 1300 BP using our 'best-fit' model parameters obtained from independent calibration. Shown are also ice cap extent in 1995 AD (black thick line) and corresponding drainage basins for outlet glaciers Rembesdalskåka (SW) and Midtdalsbreen (NE; black thin lines)




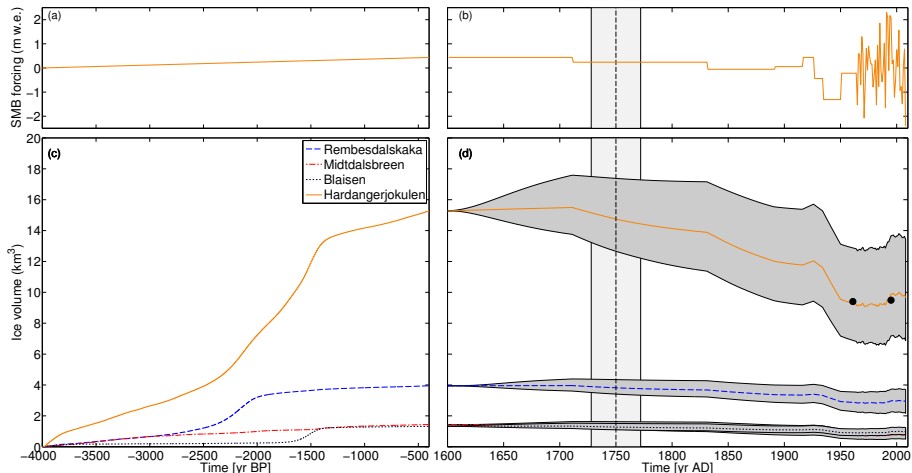

**Figure 6.** (a) Mass balance forcing for mid- to late Holocene (spinup period), and for (b) 1600 - 2008 AD. (c) Ice volume response for mid- to late Holocene and for (d) 1600 - 2008 AD using an ensemble of sliding and deformation parameter combinations (dark shading) and our 'best-fit' combination obtained from independent calibration. Colors represent different outlet glaciers and the whole ice cap. The LIA maximum, as dated at Mitdalsbreen (dashed line), and its temporal uncertainties (light shading) is also shown, as well as ice volume observations from 1961 and 1995 (black dots). For details, see text.

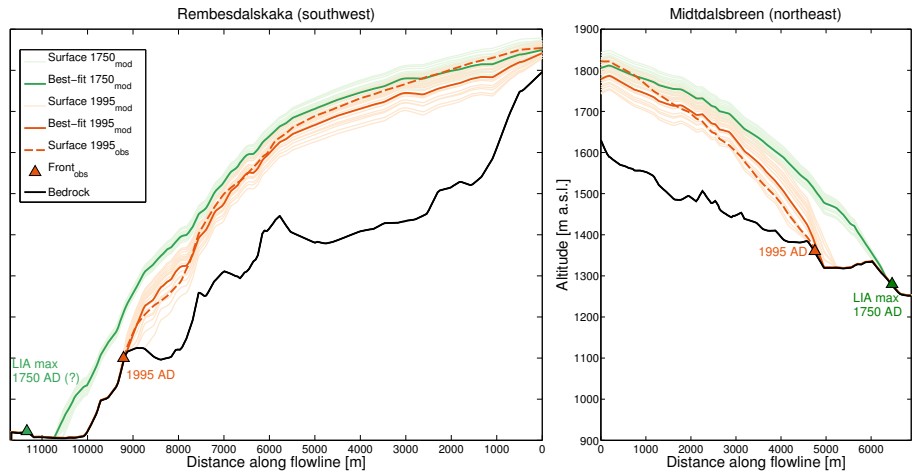

**Figure 7.** Modelled surfaces for 1750 (light green) and 1995 AD (light orange) for Rembesdalskåka and Midtdalsbreen, using an ensemble of different dynamical parameter combinations. Modelled surface using our 'best-fit' parameter combination is also shown for 1750 (green) and 1995 (orange), as well as observed surface in 1995 (dashed orange). Outlet front positions as known from dated (Midtdalsbreen) and assumed contemporary (i.e. not dated; Rembesdalskåka) terminal moraines are indicated with triangles. Note that the top of Rembesdalskåka (Hardangerjøkulen's summit) does not coincide with the top of Midtdalsbreen's flowline (Fig. 9e).




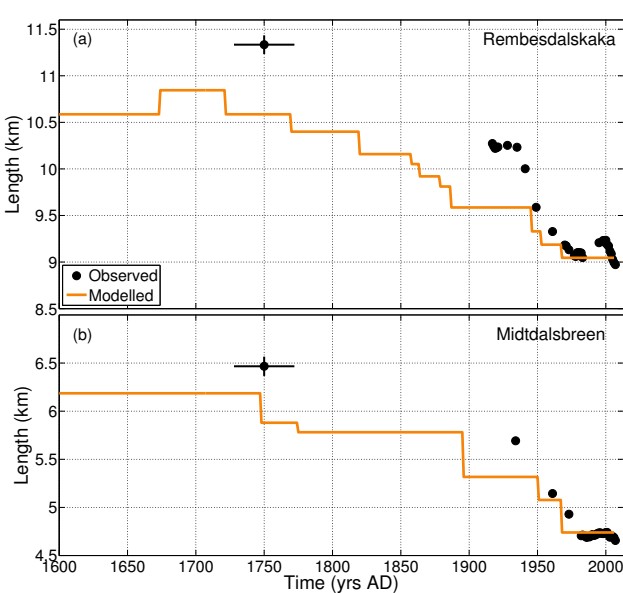

**Figure 8.** Modelled and observed length of outlet glaciers (a) Rembesdalskåka and (b) Midtdalsbreen. Temporal uncertainty for 1750 is indicated based on a 10 % age error (Innes, 1986) on the dated moraine at Midtdalsbreen (Andersen and Sollid, 1971), and assuming that the Rembesdalskåka moraine is contemporary. Uncertainties in measured lengths in the 1900s and 2000s are smaller than the marker size.





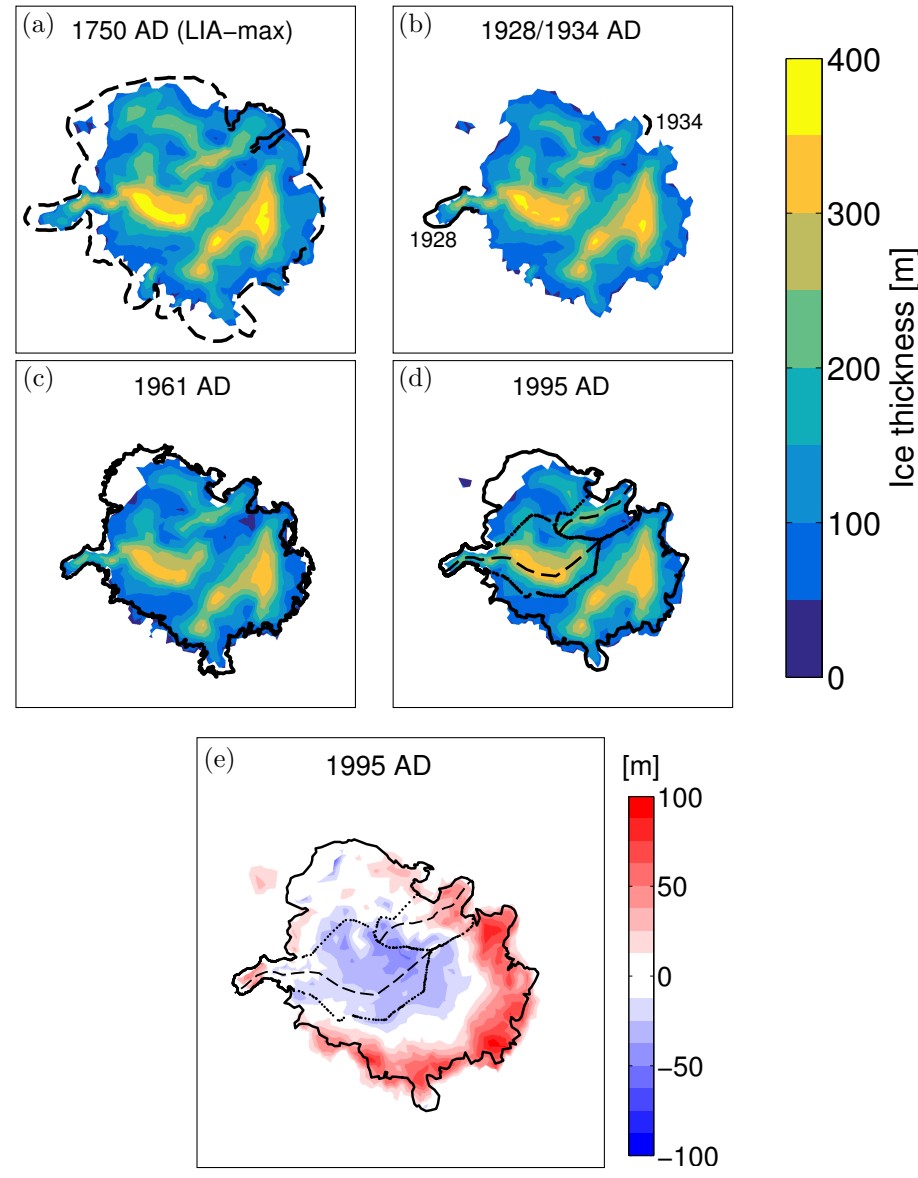

**Figure 9.** Modelled ice thickness of Hardangerjøkulen in (a) 1750, (b) 1928, (c) 1961 and (d) 1995 AD. Shown is also the difference between modelled and observed surface in 1995 (e), where positive (negative) values indicate that the model overestimates (underestimates) surface elevation. Observed ice cap extents (Andersen and Sollid (1971); Sollid and Bjørkenes (1978); A.Nesje, pers. comm; H. Elvehøy, pers. comm; Cryoclim.net/NVE) for corresponding years are shown where available. For 1750, assumed LIA extent from geomorphological evidence (dashed line) and dated LIA extent (solid line) is shown. For 1928/1934, the modelled thickness displayed is for 1928, though the observed front shown for Mitdalsbreen is from 1934. Drainage basins and flowlines of Rembesdalskåka and Midtdalsbreen are shown for 1995.



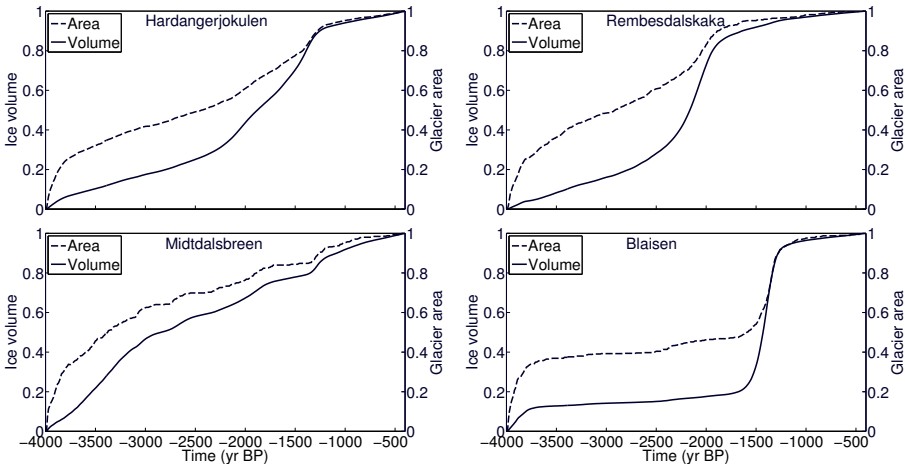

**Figure 10.** Simulated ice volume and area evolution for (a) Hardangerjøkulen, and the outlet glaciers (b) Rembesdalskåka, (c) Midtdalsbreen, and (d) Blåisen, from 4000 to 400 BP (1600 AD). Quantities are non-dimensionalized relative to final volume and area in year 1600 AD, respectively.

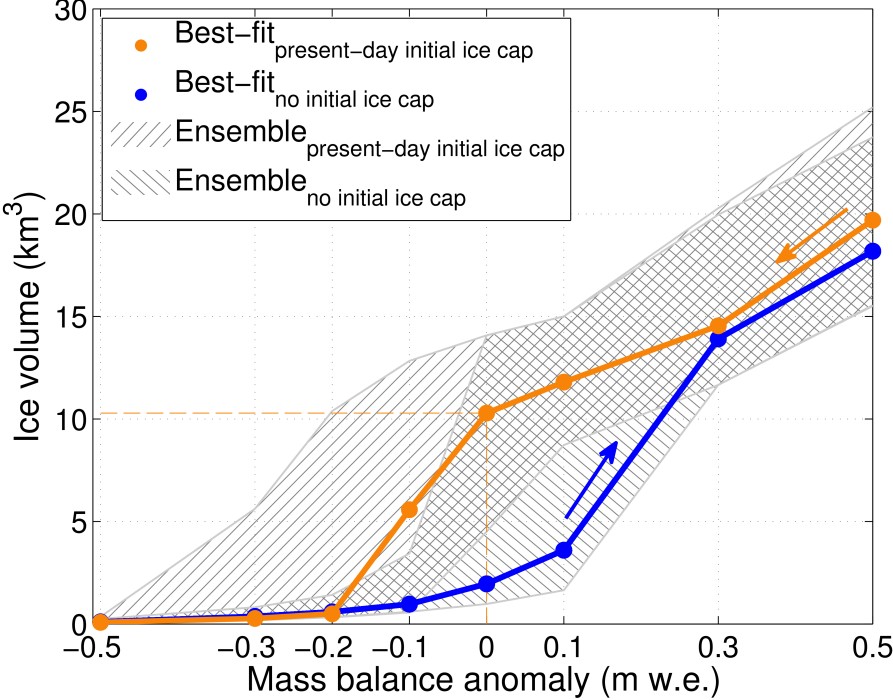

**Figure 11.** Steady-state ice volumes reached using step perturbations of the 1963-2007 mass balance, using an ensemble of dynamical parameter combinations, starting from the present-day ice cap and ice-free conditions.





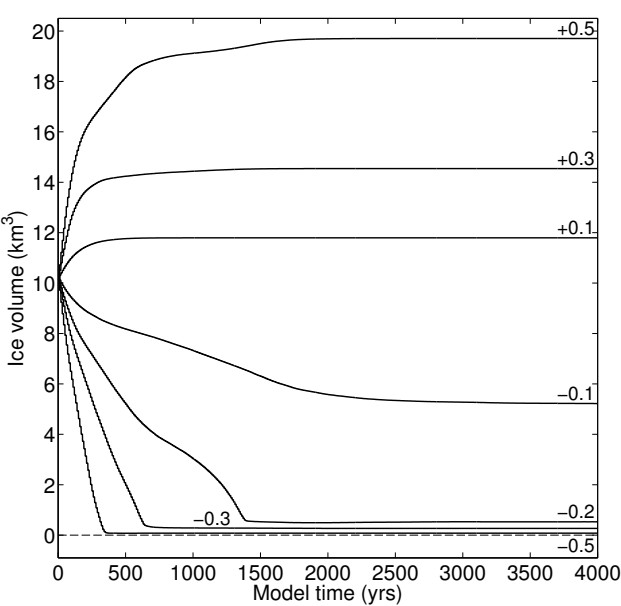

**Figure 12.** Ice volume evolution for selected mass balance perturbations (-0.5 to 0.5 m w.e.) relative to the mean mass balance 1963-2007, using our 'best-fit' dynamical parameter combination. A mass balance anomaly of -0.2 m w.e. is added for greater detail of when Hardangerjøkulen disappears.