# Peer review of "Simulating the evolution of Hardangerjøkulen ice cap in southern Norway since the mid-Holocene and its sensitivity to climate change"

_The Cryosphere, 2016_

## Referee Comment (RC1) · Anonymous Referee #1 · 23 May 2016

**General comments**

In this manuscript Åkesson and co-authors simulate the build-up of the Hardangerjøkulen ice cap (Norway) from the Mid-Holocene (4000 years ago, when there was no ice cap) to the present-day by coupling a SIA model to a simple elevation dependent mass balance model. At first a mass balance forcing based on climate reconstructions is used (Holocene), after which a switch is made a mass balance forcing based on geomorphological evidence (LIA to 1968) and finally direct surface mass balance measurements are used (1968 to present-day). This setup, with a focus on the long-term evolution of the ice cap, is interesting to get an insight in the dynamics of this ice cap and the important role of the surface mass balance (SMB) and its feedback with elevation. However, the authors do not really dig into these concepts and most of the descriptions are too site specific. Despite some attempts to make a few generalizations, the research and concepts presented here are rather trivial and no new concepts are introduced. A few interesting elements / possible points of research focus are mentioned, but then usually a reference is made to 'potential future work' / 'behind the scope of this research' and these not further elaborated.

Quite a lot of comparisons with other studies are made (often for totally different settings, which is not always appropriate) to typically conclude that similar findings are found. Moreover a lot of statements and passages are simply not supported by the results presented, which is for instance the case for the parts on ice dynamics and the comparisons between the shallow ice approximation (SIA) and more complex solutions (Full-Stokes (FS) / Higher-Order (HO)) (see also my more specific contents). I also have some strong reservations concerning some interpretations, mainly those relying on the (too) simple surface mass balance (SMB) parameterization. Furthermore the structure of the manuscript is often difficult to follow with sections in which comparisons with other studies are made, but also comparisons between earlier studies on Hardangerjøkulen and the literature. A lot of sections could be reduced, many repetitions could be avoided and the writing style can be improved.

Under this form the paper lacks scientific novelty and many of the descriptions are very general and imprecise. Some of the methodology may have to be rethought, which is especially the case for the surface mass balance, which almost fully determines the build-up and is highly uncertain. More detailed analysis and other experiments, which allow for some generalizations (i.e. findings which are less site specific), are needed for this research to be more relevant to the scientific community.

**Specific comments**

Abstract:
- First paragraph (l. 1-4, p.1): do you need this in abstract? Quite long abstract, so would consider removing this.
- l. 11: "given a linear climate forcing": the forcing was in reality not linear. You impose this. Could change this to: "Under a linear…"
- l. 13: "intriguing": this is a scientific text, something cannot be "intriguing": there is a reason behind it. Rather opt for "remarkable".
- l.16-17: in- and out-of-phase: not clear here. One has to read the manuscript to understand. Would reformulate this.
- l. 18: canonical: what does this mean?
- l. 19: "we provide new insights…" → would not formulate it this way. Let the reader decide whether he thinks it is new. To me most findings are site specific and there are little to no new insights on the long-term dynamics response of ice caps (e.g.1: the role of SMB-elevation feedback is something that has been analyzed far more in-depth and from a conceptual point of view (see my comments further); e.g.2: the fact that growth is not symmetrical and linear despite the linear forcing is also rather trivial)
- l. 21: close to observations: of course, because this is partly imposed.

Introduction:
- l.3-4: make reference to the new study by Huss and Hock (2015) here, which is the first to model all glaciers and ice caps explicitly.
- l.5-6: reference(s)?
- l.7: do not understand. GICs response essential because ice sheets are slow? (contribution ice sheets also important in next century)
- l.8: 170000 GICs: reference for number?
- l.12-17: "For comparison… into the physics operating on these time scales": strange passage. How is this related to the rest of intro?
- l.18: omit "so-called": they are Full-Stokes models.
- l.18: also add a reference to Jouvet et al. (2009) here. Far more relevant than two others given the fact that you consider a small ice mass. Study of Jouvet et al. (2009) was first to really apply FS on glacier for time dependent evolution.
- l.20: "simpler models are generally preferred": why so? Do not agree. Must make sure that you have a certain detail in data to justify the use of complex (HO/FS) model, but if this is the case and if you have the resources to do so: more complex model is more interesting. At several points in the paper the difference between SIA and FS is minimized in your interpretation: but do not rely on your results to do this, be careful. Differences can be quite large, especially in your fast flowing steep outlet glaciers.
- In this study: would have been interesting to make comparison with a more complex model, especially given the fact that you work with a model (ISSM) where this can be done! Run of 4000 years with HO

model with resolution 200-500 m is definitely feasible, especially given the very small extent of the ice cap (compared to ice sheets).

- l.22: simple models are needed to do extensive 'ensemble experiments'. Has been done in a far more elaborate and precise way by others, in a computationally heavier setup: e.g. have a close look at the recent study by Ziemen et al. (2016) (much larger domain, over the entire Alaskan Ice Field, and with more complex model, especially when it comes to the SMB), which analyses in a very nice and in depth way the effect of many parameters (not only related to ice flow and sliding)
- l.24-27: you mention centuries to millennia when it comes to response time. And one of the reasons for you to study the last 4000 years is related to the long response time of the ice cap. The long-term dynamics are important, but also the shorter time scales matter. If you apply a strong warming during several decades, the long-term evolution will quickly be altered and especially the outlet glaciers (which are quite central in your story) will react to this. Would also mention the decadal time scale here (which you mention later, in your ice flow model description, p.7, l.1-3) and some related studies (e.g. Leysinger Vieli and Gudmundsson, 2004; Raper and Braithwaite, 2009; Zekollari and Huybrechts, 2015)
- l.26: which studies? Should make a reference here.
- l.29: "carry out an extensive evaluation". Do not agree. See also my comment earlier and reference to the work of Ziemen et al. (2016).
- l. 26-29: in the end this is a passage that summarizes why "your work is better than others". Be careful with this, especially given the fact that the setup is not so unique (other long-term studies exist) and the analyses are not so in-depth (again: Ziemen et al. (2016): here the calibration is also not 'lost' (l.30))
- l. 32-33: "by considering the underlying bed topography": of course: otherwise you do not have the ice cap geometry and cannot do any modelling + the uncertainty is very large and many areas without measurements. "interacting ice dynamics": do almost not have any information about this (especially when it comes to basal sliding, a process which is discussed elaborately in your manuscript)
- p3, l.1: "model stategy": strange formulation. Rather use "metholodogy"

Section 2:
- Strange sequence: present-day → LIA → Holocene: would re-arrange this.

Section 2.1.1:
- l.9: Present-day: when is this? 2012? Quickly changes under present-day conditions. Otherwise use "about" to qualify this.
- Give a lot of info about Rembesdalskaka: what about the other outlet glaciers?

Section 2.1.2:
- Which DEM is used (needed to reconstruct the bedrock elevation)? Is this the one you mention later in section 3.2.2
- l.27-29: need interpolation for areas with small surface slope → is this only at ice divide and ice ridges. Or also in other locations? Be more specific.
- l.29-30: continuous decrease in ice thickness: towards the edge? Not fully clear, could elaborate on this.

Section 2.1.3:
- Beginning (l. 2-6): jump from one time period to another. Consider reorganizing this.
- l.7: "both outlet glaciers". There's more than two, confusing → "The two outlet glaciers considered.."

Section 2.2
- Again a strange sequence: present-day → past (Holocene + LIA) → present-day

Section 2.2.1:
- Second paragraph (l.26-30): discuss precipitation different locations and all of a sudden in last sentence a mean annual temperature is mentioned. Not related to this. Would omit this or start with new sentence in which the temperature is mentioned (also for other sites?).

Section 2.2.2:
- l.4: "is documented" → when formulated like this seems that there was someone 4000 years ago who saw this and wrote this down. Not the case. Would for instance use "is reconstructed".
- l.7: "unfavourable conditions": what is favourable/unfavourable for an ice cap? Unfavourable conditions for growth? Consider reformulating this, potentially as a function of SMB.

Section 2.2.3:
- l.19-20: SMB: 45 mass balance years. How do you define the SMB years? Not sure, but period 1963-2007: would in first instance interpret this as 44 years.
- SMB: decrease at highest altitudes. Is this decrease really so strong? Any references to other glaciers where a similar decrease is measured? Explanation: by snow redistribution (l. 21-23): is this the only mechanism? No correlation to temperature (cf. Clausius-Clapeyron) or any other explanation?
- Last sentence: approximated by second-order polynomial vs. in caption of the figure that illustrates this (figure 2): third-order polynomial? Which one is it?

Section 2.3.1:

- p.6, l. 2: first you say that the ice cap can considered as temperate (i.e. all ice at pressure melting point) and in next sentence you mention an outlet glacier to be cold-based (i.e. ice cap is polythermal and not temperate). Not consistent. Also not very clear what has been measured and what not.

Section 2.3.2:

- Very large range for velocities for lower ablation area of Midtdalsbreen: 4-40 m a$^{-1}$: the upper part of this range is even faster than the values that you mention further for around the ELA (33 m a$^{-1}$): is this really the case? Could be due to local topography/sliding/…, but otherwise would expect higher velocities around the ELA.

Section 3.1:

- Not fully sure about the formulation of the SIA. Typically explained more as a function of (glacier) width vs. ice thickness. What do you exactly mean by 'typical glacier length' (l.24)? How do you determine the 'characteristic horizontal scale' (l.29) for your ice cap to be 4-8 km (and the 'characteristic ice thickness to be around 200 m' (l.29)?)
- As I indicated before, given the model you use, a comparison between SIA and HO would have been interesting (and computationally feasible)
- Would recommend to also have a look at recent paper by Kirchner et al. (2016) who review in-depth the differences between models of different complexities for longer time scales. Interesting elements that you could (/should?) add when discussing the SIA / HO-FS differences (not only here, also for other parts in text)

Section 3.1.1:

- Be consistent in formulation with $\tau, \tau_b, \tau_d, \bar{u}_d, u_b, \bar{u}, u$, which is not the case at this point.

Section 3.1.3:

- l.22-24: really need the lower resolution? Would expect higher resolution to be computationally feasible. If opt for low resolution, would do (one) higher-resolution run for comparison also.
- l.25: need such a small time step?

Section 3.2.1:

- l.29-30: repetition (+ see earlier comment: are this 44 or 45 years of measurements?)
- SMB forcing: very simple. Not sure about applicability for other periods in time. Cannot catch many processes that are important and probably very different under other climatic conditions (changes in albedo, changes in refreezing,…etc.)
- p.9, l.3-5: elaborate. Not clear at this point.

Section 3.2.2:
- Rate factor does not only depend on ice temperature. Important, but not the sole parameter. This is for instance clear from the fact that a wide range of rate factors is used for temperate glaciers, while the temperature is always at the pressure melting point. In your discussion and rationale the focus is too much on temperatures, be careful. l.21: "corresponding to ice temperatures" → "roughly corresponding to ice temperatures".
- l.30: "Based on figure 3": cannot base yourself on figure to conclude something. You base yourself on the experiments (their outcome) and the figure illustrates this.

Section 3.2.3:
- Again start with a repetition: overlap with section 2.2.2: should re-organize this to make text more consistent.
- l. 19: "adds additional uncertainty and unnecessary complexity": be more specific. Not sure some additional complexity is unnecessary, could very well be needed to capture some processes…

Section 3.2.4:
- Last sentence (l.1-2, p.11): repeat yourself again. Would remove this.

Section 4.1:
- l.5: again a repetition.
- l.6-7: you "demonstrate" that growth is non-linear. Of course, this is not an idealized setting, so rather trivial that growth is non-linear. Is this really "demonstrating " something? Lines that follow: long part to say little.

Section 4.2.1:
- l.28: start with another repetition.
- l.30-31: have a very large spread. Of course, large ensemble, most are wrong (too stiff/slow or too viscous/fast): the range mentioned depends fully on the size of your ensemble and per se does not mean anything.

Section 4.2.2:
- Very descriptive, chaotic and lacks structure. Should reorganize this and be more specific (to-the-point) to be clearer.

Section 5.1:
- l.30-31: "this is not surprising" → would reformulate this.
- First paragraph: discussion about (basal) velocities: have very little information (especially when it comes to basal velocities) (as you mention yourself) → discussion is not really relevant.
- l.6-11: Rate factor is not only related to temperature (see earlier comment). → l.14: "corresponding to -3°C": directly relating to temperature is probably not relevant/correct.

- l.20-26: weak description. Many words to say little. In the end you say: if fast → thin / if slow/stiff → thick
- l.30 (p.13) → l.2 (p.14): mention something interesting. Would do this here. At this point the manuscript introduces a model and a (pretty straightforward) calibration/validation (and the evolution for this specific ice cap): what is the added value of this study compared to earlier studies?

Section 5.2:
- Long section about sliding: do almost not have any information. Based on your modelling → cannot really learn anything new about sliding for this ice cap. Results are simply related to your model setup and in the end your finding (which you mention further: that a lot of different combinations for your rate factor and sliding parameter are possible) is logical (as both flow and sliding have similar spatial patterns in your setup) and this was already demonstrated in earlier studies.
- Comparison with other studies on ice sheets. Is this relevant? Totally different setting, other mechanisms for water to reach the bed (/being locally produced).
- l.18:"It is therefore not surprising" → change
- l.26-29: relationship sliding and geometry: from theoretical perspective. This is not a "finding" from your study..
- l.28: "Thus, for whatever the cause,.." → If you want to know the cause: have a look into ice flow theory.. + not kind of language expected in scientific text ("for whatever the cause"..)
- p.15, l.3-4: indeed. A whole section to say very little..

Section 5.3:
- l.6-9: repeat yourself.
- l.10-13: SMB vs. elevation: too simple here. What about albedo, refreezing and for instance insolation (expect very different SMB vs. elevation for a surface oriented to the South and one oriented to the North…)
- l.19-20: what do you mean? Be more specific.
- l.23-24: indeed. Could this not be done?
- l.31-33: snow redistribution. Could indeed have an effect. But probably smaller effect than the large errors induced by your other approximations.
- p.16, l.3-8: not convinced that this error is that large compared to the magnitude of errors induced by your simple modelling..
- l.9: "works well": not sure..
- l.13-17: of course, so would need albedo in model! Does not have to be a very complex model where a lot of data is needed for validation/calibration (e.g. model solving the full energy balance): this can be done in a rather simple way, but which is very effective (e.g. PDD model, T index model, simple energy balance model,…) (e.g. Braithwaite, 1995; Hock, 2003; Oerlemans, 2001)

- l.21-23: Holocene changes in climate are strongly influenced by changes insolation, so this should be taken into account. Could be done with simple parameterization also.

Section 5.4:
- Discussion on ice dynamics, while you do not really have the material to discuss this. This is mostly a reference to the literature. A pity, given the fact that your model can be run in HO and a comparison can be made…
- p.17, l.1-4: you discuss the effect of sliding and the deterioration of the SIA as this increases. Is indeed true. Then say that because you do not necessary have information → cannot draw conclusions. This is true, but I think that the main reason why you cannot draw conclusions is simply because you do not have a 'reference run' (a HO/FS run) to compare to.
- l.5-13: this is not a discussion of your results.

Section 5.5:
- l.23: effect SIA. Of course true, but the effect of SIA/HO-FS is very limited compared to other errors and uncertainties. Over Holocene timescale the SMB (where uncertainties are large) will have much larger effect than dynamics on the evolution/growth.

Section 5.6:
- l.4-9: growth → very descriptive and site specific. What is added value for reader?
- l.21-22: "this asymmetry illustrates that proxy records representing different parts of an ice cap may lead to substantially different conclusions about ice cap size through time" → of course. Rather trivial.
- l.23-32: long passage with little information.
- p.19, l.1-27: many words about response time to in the end say very little. Do not have experiments to elaborate on this. Could spend a few words on this, but not whole section.
- l.28 → p.20, l.1-2: not sure that your results support this. Rather speculative.

Section 5.7:
- l.7: effect proglacial lake. Can have an effect, but expect this again to be much smaller than other model uncertainties.
- l.14: "in our view a step forward": not sure. Even if would be the case, you should maybe not write this down and let the reader decide for himself whether he thinks this is new/novel/better than methodology applied in other studies. First focus should be a carefully calibrated/validated and robust setup, supported by field data, and not sure whether this is the case in this study.

Section 5.8:
- l.25: you "show" that ice cap is very sensitive to change in climatic conditions. Trivial: of course, it is an ice cap. Importance SMB-elevation feedback. Has been analyzed in (far greater) depth and from theoretical point in the past. Have a look at some of the 'classic' papers on this (Lee and North, 1995; Mahaffy, 1976; North, 1984).
- p.21, l.1-4: again rather trivial. What's new about this finding?
- l.9: 750 years to disappear. Too precise. Would change this to "around 750 years"
- "As evident from Collins et al. (2013), we expect a warming scenario": strange formulation.
- l.11-21: do not really discuss your own results, not based on your simulations.
- l.22-28: what's new?

Conclusions:
- Start from ice-free in Holocene: do you also get this if would start simulations earlier and force with a palaeoclimatic record? Would be an interesting experiment..
- p.22, l.3-6: this is not something new. Not a finding from this study.
- l. 9-14: SMB-elevation feedback exists for ice cap. You show this, but do not really add anything new to the theory related to this.
- l.15-17 + l. 24-26: site specific → what is the more general interest?
- l.27-31: strange way to end your conclusion..

Figures:
- Nice and clear figures in general.

**References**

Braithwaite, R. J.: Positive degree-day factors for ablation on the Greenland ice sheet studied by energy-balance modelling, J. Glaciol., 41(137), 133–160, 1995.

Hock, R.: Temperature index melt modelling in mountain areas, J. Hydrol., 282(1-4), 104–115, doi:10.1016/S0022-1694(03)00257-9, 2003.

Huss, M. and Hock, R.: A new model for global glacier change and sea-level rise, Front. Earth Sci., 3(September), 1–22, doi:10.3389/feart.2015.00054, 2015.

Jouvet, G., Huss, M., Blatter, H., Picasso, M. and Rappaz, J.: Numerical simulation of Rhonegletscher from 1874 to 2100, J. Comput. Phys., 228(17), 6426–6439, doi:10.1016/j.jcp.2009.05.033, 2009.

Kirchner, N., Ahlkrona, J., Gowan, E. J., Lötstedt, P., Lea, J. M., Noormets, R., von Sydow, L., Dowdeswell, J. A. and Benham, T.: Shallow ice approximation, second order shallow ice approximation, and full Stokes models: A discussion of their roles in palaeo-ice sheet modelling and development, Quat. Sci. Rev., 135, 103–114, doi:10.1016/j.quascirev.2016.01.013, 2016.

Lee, W. and North, G.: Small ice cap instability in the presence of fluctuations, Clim. Dyn., 1995.

Leysinger Vieli, G. J.-M. C. and Gudmundsson, G. H.: On estimating length fluctuations of glaciers caused by changes in climatic forcing, J. Geophys. Res., 109(F1), F01007, doi:10.1029/2003JF000027, 2004.

Mahaffy, M. W.: A three-dimensional numerical model of ice sheets: Tests on the Barnes Ice Cap, Northwest Territories, J. Geophys. Res., 81(6), 1059–1066, doi:10.1029/JC081i006p01059, 1976.

North, G. R.: The Small Ice Cap Instability in Diffusive Climate Models, J. Atmos. Sci., 41(23), 3390—3395, doi:10.1175/1520-0469, 1984.

Oerlemans, J.: Glaciers and Climate Change, Balkema., 2001.

Raper, S. and Braithwaite, R.: Glacier volume response time and its links to climate and topography based on a conceptual model of glacier hypsometry, Cryosph., 183–194, 2009.

Zekollari, H. and Huybrechts, P.: On the climate–geometry imbalance, response time and volume–area scaling of an alpine glacier: insights from a 3-D flow model applied to Vadret da Morteratsch, Switzerland, Ann. Glaciol., 56(70), 51–62, doi:10.3189/2015AoG70A921, 2015.

Ziemen, F., Hock, R., Aschwanden, A., Khroulev, C., Kienholz, C., Melkonian, A. K. and Zhang, J.: Modeling the evolution of the Juneau Icefield between 1971 and 2100 using the Parallel Ice Sheet Model (PISM), J. Glaciol., 2016.

---

## Referee Comment (RC2) · Anonymous Referee #2 · 30 May 2016

**General Statement:**

The paper presents some modelling results, which concern the growth and the retreat of the Hardangerjokulen ice cap, South Norway, from the mid-Holocene to the present-day. To do so, the authors have used the Ice Sheet System Model to simulate the dynamical evolution of the ice cap. The model accounts for internal ice dynamics (SIA), linear basal sliding and surface mass balance. Dynamical parameters (sliding and shearing of ice) are first calibrated at present-day, and secondly in transient runs. The transient simulations indicate an asymmetry between southwest and northeast section during both advancing and retreating stages.

This study is in line with number of previous papers (referenced in the manuscript),

which present modelling results for a specific glacial area, and compare the results to field evidence. I have no doubt that the methodology can be successfully applied to gain insights about the chronology of the advances and retreat of this particular ice cap and complement the geormophological information already available. Unfortunately, I find the present manuscript hard to follow so that the main achievement of the paper is somehow hidden. One reason to explain that is: the paper does not follow any clear continuous line, which should bring the reader from the original investigated problem to some final conclusions. The paper spends a lot of sentences to discuss things of little importance/originality or already debated many times, and this strongly harms the overall reading. Unfortunately, the most interesting results arrive at the end of the paper, so that it is likely that most of readers won't reach this point (being discouraged by too many unnecessary discussions). In addition, the paper shows number of inaccurate/inappropriate/awkward sentences, which harm the overall argumentation (see some examples below). I believe that the paper must be rewritten before to be reconsidered for publication. This including a substantial shortening (removing unnecessary/distracting parts, and better emphasizing the main outcomes). I hope that my next suggestions will help the authors to achieve this task.

**Major concerns:**

- Section 5.2 is a typical example of section, which really slow down the reading because of a lack of originality and importance for the present study. I would recommend to remove it, or to keep the most relevant information (maybe to be merged with Section 5.1). I believe that Section 5.3 could be more efficiently and more concisely rewritten. More generally, the whole Section 5 should be "optimized".

- The manuscript contains hazardous/inaccurate statements, and sometimes awkward/dangerous assessments. However, I believe this is more unfortunate formulations rather than misunderstandings by the authors. For instance:

- p. 7: "Where bed and surface topography is complex, lateral drag and longitudinal stress gradients may become important. Still, the SIA has proven accurate in representing glacier length and volume fluctuations on decadal and longer time scales." is more confusing (and even contradictory) than useful.

- l.23 p. 8: "Even though our surface digital elevation model (DEM) has higher resolution than this (100 m), we choose the highest mesh resolution to be 200 m, since this is more in line with the assumption of the SIA" In what the mesh size and the physical model (here SIA) are connected?

- "SIA is viable to use if interests are climatic rather than ice dynamics." should be more accurate.

- "By investigating a small valley glacier in the Canadian Rocky Mountains and neglecting basal sliding, Adhikari and Marshall (2013) suggested that SIA performs well in less 'dynamic' settings, while the results compared to HO/FS diverge for more 'dynamic' situations." is a striking example: of course, if there is less dynamics, then the errors related to the dynamics gets less visible!

- l.1 p.17: "It is challenging to assess how much sliding there could be before SIA validity deteriorates, but it likely depends on the climatic and glaciological setting. " is inaccurate

- l. 23 p. 17 "we are aware of the limitation ... Therefore, the actual rate of advance may differ...". The actual rate of advance may differ because you are aware of the limitation of the SIA?

- The paper is poorly structured. Some information come repetitively in the paper (as the justification of using the SIA). The discussion (Section 5) looks more like a list of items without connections between the subsections. The last paragraphs of each subsection of Section 5 state some recommendations for future stud-

ies. I think this is not the right place, such statements should rather appear in a dedicated "perspective" closing section.

- The dynamical model is essentially based on the most simple existing model, namely the Shallow Ice Approximation. Even if this is a surprising choice (regarding to the capabilities of ISSM, and other higher order models freely available nowadays), I find unnecessary to describe in details the well known SIA so that Section 3.1 can be strongly shortened. In addition, there are several clumsy attempts to justify the use of the SIA throughout the whole paper. The uncertainty due to mechanical simplifications cannot be quantified since no comparative tests are done with higher order solutions. As a consequence, I don't see the point of discussing so much in details this assumption, while a simple referencing to previous comparative studies (e.g. Lemeur and al) would be enough.

- All what concerns the calibration to ice flow and sliding parameters should be strongly shortened since this problem has been presented many times so that only the result matters. Also, I am not really convinced by the "best-fit" pair of parameters, which is chosen among all those which minimize equally the RMSE. If I understand correctly, the 'best-fit' parameters are chosen according to the temperature of the equivalent rate (Arrhenius) factor A, which should correspond to temperate ice. This is a very weak argument, which cannot be used to constrain A. Most of ice flow models are tuned through enhancement factors, this indicates that one cannot rely directly on the exponential formula for A(T) given in (Cuffey and Paterson, 2010, p. 73). Say differently, the formula works in a relative way (after tunning), but not in a absolute way.

- I don't see in what the spatial asymmetry is intriguing or unexpected since you mention several times that precipitation are asymmetric (west-east precipitation gradient).

- Figure 11 and 12 are to me the most interesting results of the paper, so that

they deserve to be better highlighted. However, the authors should clarify in what these results are different from the Figures 6.7 and 6.8 of (Giesen, 2009, PhD dissertation), which is already based on the SIA.

**Specific comments:**

- Abstract and later: I don't understand why you say that your model is 2D. The SIA provides a 3D velocity field. To me, your model is 3D.

- Abstract and later: You often emphasise the capabilities of ISSM to perform mesh refinements, but you never say what does it brings to the study. If this is not relevant, it should be in the abstract.

- Eq. (1): I don't think it is necessary to repeat the formula used to reconstruct the ice thickness (or equivalently the bed). Referencing would be enough.

- l. 12 p.4: I don't think that the acronym NVE was defined so far.

- l. 4 p.5: "At c. 4000 BP" and many other places in the text: What "c." stands for?

- Eq. (4) dot is missing at the end. Punctuation (coma and dots) is sometimes missing in your equations.

- l.25 p.7: "SIA" must be "The SIA".

- l.7 p.8 "ISSM has capacities ..." this is useless information since you don't use this capability.

- Eq. (7) $u$ should be $\bar{u}$.

- l. 13 p. 8: is the unit of $M$ not m a$^{-1}$? mass balance rate should rather be annual mass balance?

- l. 31 p. 12: "both depend on driving stress", ok but I think one can explain much more easily why several pairs of $A$ and $\beta$ gives similar RMSE: one can reduce sliding and increase shear of ice while keeping same surface velocities.

- l. 10 p. 13. As I said before, "We therefore exclude ....". I don't think you can use this argument to eliminate pairs of parameters. But instead, it sounds more reasonable to keep going with several pairs $(A, \beta)$ which would be a set of "best-fit" parameters.

- l. 20 p.13 : You havn't defined $T_{ice}$.

- l. 20 - l.27 p.13 : This paragraph is especially laborious to read, and can be certainly shortened.

- l. 29 p. 13: It makes no sense to refer to numerical objects (mesh node) in this Section.

- l. 33 p.13: "By imposing ... ice masses". This sentence doesn't bring anything.

- l. 34 p. 13: "cold- to warm-based" I guess you refer to basal condition? If yes, you should formulate that more clearly.

- l. 17-18 p. 17: ", since the ice present is divided ..." I don't understand the meaning.

- You should maybe rename Section 5.3 or 5.8, since it seems (from the names) that they address the same issue.

- l. 1-4 p.21: why don't you show the results?

---

## Short Comment (SC1) · 21 Jun 2016

Sebastian Luening

luening@uni-bremen.de

While studying this manuscript I could not find any hint towards the Medieval Climate Anomaly (MCA). This is a phase which was anomalously warm in the study region and which was generally associated with a major glacier melt episode. Why is the MCA not featured in the article? The most likely driver of the MCA warming is high solar activity during this phase, requiring an adequate radiative forcing to produce an effect in the theoretical models. I strongly recommend to include a discussion on the MCA in the paper. Reference to the Little Ice Age requires an analysis of the preceding climatic event, i.e. the MCA, otherwise the discussion is incomplete.

---

## Author Comment (AC1) · 19 Aug 2016

**Response to Reviewer #1**

We would like to thank the referee for his/her thorough review with insightful and constructive comments. Several valid points are raised, which we respond to in a point-by-point manner below, our responses in blue. We are especially thankful for the reviewer's many detailed suggestions, which have improved the manuscript greatly. We have in addition made substantial efforts to improve structure and clarity.

On behalf of the authors,
Henning Åkesson

**General comments**

In this manuscript Åkesson and co-authors simulate the build-up of the Hardangerjøkulen ice cap (Norway) from the Mid-Holocene (4000 years ago, when there was no ice cap) to the present-day by coupling a SIA model to a simple elevation dependent mass balance model. At first a mass balance forcing based on climate reconstructions is used (Holocene), after which a switch is made a mass balance forcing based on geomorphological evidence (LIA to 1968) and finally direct surface mass balance measurements are used (1968 to present-day). This setup, with a focus on the long-term evolution of the ice cap, is interesting to get an insight in the dynamics of this ice cap and the important role of the surface mass balance (SMB) and its feedback with elevation. However, the authors do not really dig into these concepts and most of the descriptions are too site specific. Despite some attempts to make a few generalizations, the research and concepts presented here are rather trivial and no new concepts are introduced. A few interesting elements /possible points of research focus are mentioned, but then usually a reference is made to 'potential future work' / 'behind the scope of this research' and these not further elaborated.

We think the present setup and focus (long term reconstruction/evolution of an ice cap using transient numerical modelling) is not commonly found in the literature and that our findings have implications for reconstructions and predictions of ice caps in other regions than Hardangerjøkulen. We agree however with the reviewer in the sense that the transferability and novelty was not clear in the original manuscript.

  To improve this, we have now strengthened our focus on the SMB-elevation feedback. Originally in the Discussion, these findings have now been moved to the Results, to increase visibility. Simulations excluding the feedback have been added to Fig. 12. We do believe that the strong role of this feedback on the time scales we consider is relevant not only for Hardangerjøkulen but for studies of other maritime ice caps, e.g. in Norway, Alaska, Iceland and Patagonia, because of the similar hypsometries and mass balance regimes of these ice caps. In addition, we think that the dependency of initial conditions for ice caps (hysteresis), illustrated in Fig. 11, has not received much attention in the literature and is relevant for modelling and reconstructing paleo-ice caps and predicting future ice cap evolution. The out-of-phase variations of area and volume (Fig. 10) we find also have implications for such studies. We now further underline this study's relevance and transferability in the Introduction and have added a separate section on this in the Discussion.

Regarding "digging into" the ice dynamics, see responses below.

Quite a lot of comparisons with other studies are made (often for totally different settings, which is not always appropriate) to typically conclude that similar findings are found. Moreover a lot of statements and passages are simply not supported by the results presented, which is for instance the case for the parts on ice dynamics and the comparisons between the shallow ice approximation (SIA) and more complex solutions (Full-Stokes (FS) / Higher-Order (HO)) (see also my more specific contents).

We agree with the reviewer that the discussion of ice dynamics was not entirely appropriate for our study. Therefore, we have now rewritten these parts and refrain from making inferences about HO/FS since, as the reviewer rightly point out below, we have not done such comparative studies.

Regarding SIA/HO/FS, we do believe that there is a value in justifying our choice of SIA, especially since both reviewers suggest that a HO/FS model should be used if available.

On this topic, reviewer 1 writes in the Specific comments on the Introduction: "At several points in the paper the difference between SIA and FS is minimized in your interpretation: but do not rely on your results to do this, be careful. Differences can be quite large, especially in your fast flowing steep outlet glaciers."

In contrast, in the comments on Section 5.5, the reviewer suggests that "… the effect of SIA/HO-FS is very limited compared to other errors and uncertainties. Over Holocene timescale the SMB (where uncertainties are large) will have much larger effect than dynamics on the evolution/growth."

We are not sure where the reviewer stands here, but we agree with the second comment that SIA/HO-FS differences are likely small on the time scales we are interested in. In general, the SIA is considerably cheaper and allows for ensemble and longer time scale studies. We believe that HO/FS is unnecessary for the long time scales studied here, on this ice cap lacking areas of fast flow. This is in line with what previous studies have shown (referenced in the manuscript). Even if we had attempted a SIA/HO comparison within ISSM, it would not have been straightforward. The problem is that the parameterization of the basal friction for SIA and HO is different in ISSM; SIA parameterizes basal velocities and HO parameterizes basal stress. We therefore do not think a SIA/HO twin simulation would be informative.

I also have some strong reservations concerning some interpretations, mainly those relying on the (too) simple surface mass balance (SMB) parameterization.

See our response on the physical basis of our SMB parameterization in Specific comments, Section 5.3, below.

Furthermore the structure of the manuscript is often difficult to follow with sections in which comparisons with other studies are made, but also comparisons between earlier studies on Hardangerjøkulen and the literature. A lot of sections could be reduced, many repetitions could be avoided and the writing style can be improved.

We agree with the reviewer that the original manuscript could have been clearer and more concise. Substantial efforts have therefore gone into restructuring the

manuscript. We have now reworked the Abstract, and completely rewritten the Introduction, clearly stating the scientific questions we address and the reasons for doing so. Some subsections in the Methods have been shortened. In the Discussion, subsections are now more explicitly linked together and paragraphs not following the main aims and scope of the paper have been deleted. In the Conclusion, we highlight our main findings more directly linked to the scientific aims outlined in the Introduction.

Under this form the paper lacks scientific novelty and many of the descriptions are very general and imprecise. Some of the methodology may have to be rethought, which is especially the case for the surface mass balance, which almost fully determines the build-up and is highly uncertain.
We agree that the SMB is uncertain and is crucial for the Holocene evolution. However, our aim is not to reconstruct SMB for the Holocene, but to assess the long-term dynamic response to a simple climate forcing. We agree with the reviewer that the SMB forcing is simple; we have made it so deliberately and view this as a strength rather than challenge. This because we would like to isolate the effect of bed topography/geometry/dynamics, given a simple, imposed (linear) climate forcing.
 See also our response on the physical basis of our SMB parameterization in Specific comments, Section 5.3, below.

More detailed analysis and other experiments, which allow for some generalizations (i.e. findings which are less site specific), are needed for this research to be more relevant to the scientific community.
As mentioned above, we now add a separate section on transferability/applicability of our results the Discussion. We have analysed volume and area evolution further, and illustrate this in a new figure, relevant for volume-area scaling (in addition to the existing Fig. 10).

**Specific comments**
Abstract:
- First paragraph (l. 1-4, p.1): do you need this in abstract? Quite long abstract, so would consider removing this.
  We agree that the abstract was long and lacked focus. We have shortened the "motivation" part of the abstract to one sentence.

- l. 11: "given a linear climate forcing": the forcing was in reality not linear. You impose this. Could change this to: "Under a linear…"
  Indeed. Changed.

- l. 13: "intriguing": this is a scientific text, something cannot be "intriguing": there is a reason behind it. Rather opt for "remarkable".
  True, though we prefer to change to "distinct".

- l.16-17: in- and out-of-phase: not clear here. One has to read the manuscript to understand. Would reformulate this.
  Reformulated to "we find that for several outlet glaciers and indeed for

the entire ice cap, volume and area vary out-of-phase for multiple centuries during the late Holocene, and in-phase approaching the LIA."

- l. 18: canonical: what does this mean?
  With "canonical" we here mean an assumption that is commonly used/ recognized/established/prevailing.

- l. 19: "we provide new insights…" → would not formulate it this way. Let the reader decide whether he thinks it is new. To me most findings are site specific and there are little to no new insights on the long-term dynamics response of ice caps (e.g.1: the role of SMB-elevation feedback is something that has been analyzed far more in-depth and from a conceptual point of view (see my comments further); e.g.2: the fact that growth is not symmetrical and linear despite the linear forcing is also rather trivial)
  We thank the reviewer for this suggestion. This has been reformulated. Regarding the SMB-elevation feedback, we now assess its effect more extensively and have moved its place from the Discussion to Results. We think that the asymmetric/asynchronous, non-linear response to linear forcing deserves attention. It has implications for paleostudies aiming to reconstruct ice caps as well as for future predictions, and has in our view not received enough appreciation in the literature. We now highlight this further in our new section on transferability/applicability.

- l. 21: close to observations: of course, because this is partly imposed.
  It is true that we estimate the SMB forcing between 1600-1962 based on the length variations of two outlet glaciers. However, the forcing is not aggressively tuned (as pointed out in the manuscript). The close fit between modeled and observed ice cap margins in the second half of the 20th century is not a given, and shows that SMB plays a key role. We now reformulate ourselves stressing that not only calibrated lengths correspond well, but also ice cap extent in general.

Introduction:
- l.3-4: make reference to the new study by Huss and Hock (2015) here, which is the first to model all glaciers and ice caps explicitly.
  Thanks for making us aware of this study. We now cite it.

- l.5-6: reference(s)?
  Changed.

- l.7: do not understand. GICs response essential because ice sheets are slow? (contribution ice sheets also important in next century)
  This is indeed confusing. We now specify that both GICs and ice sheet contributions are important for 21st century sea level rise.

- l.8: 170000 GICs: reference for number?
  This number is now actually more than 211 000, according to the latest version of the Randolph Glacier Inventory by GLIMS (version 5.0,

www.glims.org/RGI). We now reference this.

- l.12-17: "For comparison… into the physics operating on these time scales": strange passage. How is this related to the rest of intro?
  We agree that this is not clear. Our reworked Introduction more clearly links this to our focus on long term transient modelling and its relevance/implications for glacier reconstructions.

- l.18: omit "so-called": they are Full-Stokes models.
  Done.

- l.18: also add a reference to Jouvet et al. (2009) here. Far more relevant than two others given the fact that you consider a small ice mass. Study of Jouvet et al. (2009) was first to really apply FS on glacier for time dependent evolution.
  We thank the reviewer for this suggestion and now cite Jouvet et al. (2009) here.

- l.20: "simpler models are generally preferred": why so? Do not agree. Must make sure that you have a certain detail in data to justify the use of complex (HO/FS) model, but if this is the case and if you have the resources to do so: more complex model is more interesting. At several points in the paper the difference between SIA and FS is minimized in your interpretation: but do not rely on your results to do this, be careful. Differences can be quite large, especially in your fast flowing steep outlet glaciers.
  One advantage of simpler models/SIA is given in the sentence after (l.21-22): simpler models allow for more extensive ensembles and longer runs, because they are cheaper. We acknowledge however that there are different schools of thought here. Therefore we now rephrase this, pointing out the advantages of simpler models without concluding whether they in general are "preferred", leaving it up to the reader to decide what he/she thinks.
  The reviewer also points on something else here: a certain detail in data is needed to justify the use of a HO/FS model. We do not believe that this data is available to us, nor are we focusing on the short-term variations where HO/FS possibly have an effect. We acknowledge however that the rationale behind simpler models was not clear and have now rewritten this passage.

- In this study: would have been interesting to make comparison with a more complex model, especially given the fact that you work with a model (ISSM) where this can be done! Run of 4000 years with HO model with resolution 200-500 m is definitely feasible, especially given the very small extent of the ice cap (compared to ice sheets).
  As the reviewer rightfully point out, a HO model for 4000 years is indeed feasible, even an ensemble study could be done. But we are not convinced we should justify HO/FS simulations by availability rather than applicability. As described in detail before, a SIA/HO/FS comparison is

not straightforward and we do not think it would be informative. Again, we now articulate the rationale behind our simple model more clearly.

- • l.22: simple models are needed to do extensive 'ensemble experiments'. Has been done in a far more elaborate and precise way by others, in a computationally heavier setup: e.g. have a close look at the recent study by Ziemen et al. (2016) (much larger domain, over the entire Alaskan Ice Field, and with more complex model, especially when it comes to the SMB), which analyses in a very nice and in depth way the effect of many parameters (not only related to ice flow and sliding)
  We thank the reviewer for directing us to this relevant study, which contains several interesting findings and improves our knowledge of ice fields, their outlet glaciers and how to model them. Though some longer simulations are done, the main focus of Ziemen et al. (2016) is predicting the next 100 years. This focus is quite different from ours. However, we now analyze our ensemble in more depth by detailing individual runs in Fig. 6 and further discuss the effect of the dynamical parameters. Our choice of a simple SMB profile is discussed in the Specific comments, Section 5.3, below.
  We now cite Ziemen et al. (2016) in the Introduction and Discussion.

- l.24-27: you mention centuries to millennia when it comes to response time. And one of the reasons for you to study the last 4000 years is related to the long response time of the ice cap. The long-term dynamics are important, but also the shorter time scales matter. If you apply a strong warming during several decades, the long-term evolution will quickly be altered and especially the outlet glaciers (which are quite central in your story) will react to this. Would also mention the decadal time scale here (which you mention later, in your ice flow model description, p.7, l.1-3) and some related studies (e.g. Leysinger Vieli and Gudmundsson, 2004; Raper and Braithwaite, 2009; Zekollari and Huybrechts, 2015)
  We agree with the reviewer that decadal time scales are indeed important. The references Leysinger-Vieli and Gudmundsson (2004) and Zekollari and Huybrechts (2015) are cited elsewhere in the manuscript but are now referenced together with Raper and Braithwaite (2009) also in the Introduction and ice flow model description.

- l.26: which studies? Should make a reference here.
  The references were partly given in the previous sentence. We now make it clear what "Studies" we mean.

- l.29: "carry out an extensive evaluation". Do not agree. See also my comment earlier and reference to the work of Ziemen et al. (2016).
  See our response above regarding Ziemen et al. 2016.

- l. 26-29: in the end this is a passage that summarizes why "your work is better than others". Be careful with this, especially given the fact that the setup is not so unique (other long-term studies exist) and the analyses are

not so in-depth (again: Ziemen et al. (2016): here the calibration is also not 'lost' (l.30))

As said before, we think that there are several important results in this study, and that long-term transient modeling/reconstruction studies of ice caps in general are rare. We do however agree with the reviewer that this can be made clearer in the manuscript. We have therefore completely rewritten the Introduction and added a new section in the Discussion, focusing on transferability/implications.

- l. 32-33: "by considering the underlying bed topography": of course: otherwise you do not have the ice cap geometry and cannot do any modelling + the uncertainty is very large and many areas without measurements. "interacting ice dynamics": do almost not have any information about this (especially when it comes to basal sliding, a process which is discussed elaborately in your manuscript)

We are thankful that the reviewer points out this imprecise wording. Having a bedrock DEM is indeed a prerequisite for the type of modelling we do. In contrast, a bedrock DEM is not always available for glacier reconstructions, ice volume estimates (e.g. volume-area scaling for sea level rise), or other applications. By "...glacier reconstructions can be improved by considering the underlying bed topography...", we tried to convey that studies aiming to reconstruct an ice cap or glacier through time would benefit from assessing/acknowledging/quantifying the impact of the bed topography on ice flow and mass balance, and therefore on the reconstruction itself. This follows from our finding that a spatially symmetric SMB and linear climate forcing result in a spatially asymmetric, non-linear response, whose explanation include the impacts of bed topography.

We acknowledge that the accuracy of the bed topography varies for Hardangerjøkulen, as for other ice masses, and already point this out in the Discussion (p.12, l.13; p.17, l.2; p.20, l.6).

"...interacting ice dynamics" is indeed not appropriate, we have now changed this to "surface mass balance", since the SMB-elevation feedback together with bed topography is vital to the long-term evolution reconstructions mainly are interested in.

- p3, l.1: "model stategy": strange formulation. Rather use "metholodogy"
Changed to "methodology".

**Section 2:**
- Strange sequence: present-day → LIA → Holocene: would re-arrange this.
Good suggestion, the order is now chronological.

**Section 2.1.1:**
- l.9: Present-day: when is this? 2012? Quickly changes under presentday conditions. Otherwise use "about" to qualify this.
Indeed not clear, this specific survey was in 2010, which is now stated.

- Give a lot of info about Rembesdalskaka: what about the other outlet glaciers?
  This focus reflects that SMB measurements are done on Rembesdalskåka, and nowhere else. We agree however that at least Midtdalsbreen should have been given some attention, since this is the other outlet glacier we focus on, and we have now added additional information.

**Section 2.1.2:**

- Which DEM is used (needed to reconstruct the bedrock elevation)? Is this the one you mention later in section 3.2.2
  The 1995 DEM mentioned in Section 3.2.2. is indeed what we use. This DEM is a result of several preceding surveys, mentioned in Section 2.1.2. We now specify this also here.

- l.27-29: need interpolation for areas with small surface slope → is this only at ice divide and ice ridges. Or also in other locations? Be more specific.
  The manual extrapolation (not interpolation) was required at ice ridges and divides. This is also detailed in Giesen and Oerlemans (2010), p.93. We now reformulate this more clearly.

- l.29-30: continuous decrease in ice thickness: towards the edge? Not fully clear, could elaborate on this.
  We now clarify that near ice margins (e.g. last km), instead of using Eq. (1), manual extrapolation of ice thickness measurements was needed to obtain a meaningful/smooth ice surface.

**Section 2.1.3:**

- Beginning (l. 2-6): jump from one time period to another. Consider reorganizing this.
  We aimed to describe the data chronologically (l. 1-9), and then summarize how we use it in our study (l. 11-12), but we agree that this was not easy to follow. We have rewritten this passage to obtain further clarity.

- l.7: "both outlet glaciers". There's more than two, confusing → "The two outlet glaciers considered.."
  Changed.

**Section 2.2**

- Again a strange sequence: present-day → past (Holocene + LIA) → present-day
  We have now switched to a chronological order, for consistency with the glacier data.

**Section 2.2.1:**

- Second paragraph (l.26-30): discuss precipitation different locations and all of a sudden in last sentence a mean annual temperature is mentioned.

Not related to this. Would omit this or start with new sentence in which the temperature is mentioned (also for other sites?).

Good suggestion, we now keep temperature in a separate sentence. Only precipitation is measured at Liset, which is now pointed out. Finse is the closest meteorological station and temperature does not vary as much spatially as precipitation does. We therefore think it is sufficient to mention the temperature at Finse.

**Section 2.2.2:**

- l.4: "is documented" → when formulated like this seems that there was someone 4000 years ago who saw this and wrote this down. Not the case. Would for instance use "is reconstructed".
  Changed to "is reconstructed".

- l.7: "unfavourable conditions": what is favourable/unfavourable for an ice cap? Unfavourable conditions for growth? Consider reformulating this, potentially as a function of SMB.
  Reformulated to "implying a more negative surface mass balance and thus unfavourable conditions for glacier growth"

**Section 2.2.3:**

- l.19-20: SMB: 45 mass balance years. How do you define the SMB years? Not sure, but period 1963-2007: would in first instance interpret this as 44 years.
  SMB years are defined from 1 Oct the previous year until 30 Sep in the year mentioned. 1963-2007 runs from Oct 1962 to Sep 2007, totalling 45 years.

- SMB: decrease at highest altitudes. Is this decrease really so strong? Any references to other glaciers where a similar decrease is measured? Explanation: by snow redistribution (l. 21-23): is this the only mechanism? No correlation to temperature (cf. Clausius-Clapeyron) or any other explanation?
  The change in SMB gradient at the ice cap plateau and the decrease at the highest elevations is a persistent feature of the winter mass balance. It is strongest in the years with large accumulation (see Fig. 5.3 in Giesen (2009), PhD thesis for specific winter balance profiles). Of the other Norwegian glaciers with winter mass balance measurements, only Engabreen in northern Norway also has a decreasing mass balance at the highest elevations, although less pronounced. What may be of influence, is that Rembesdalskåka is flowing due west, while other Norwegian ice cap outlet glaciers with observations have no or a smaller westward component. Globally, winter mass balance profiles are only available for a small number of glaciers and we are not aware of any other ice cap outlet glaciers that show a similar decrease. The suggestion by the reviewer that Clausius-Clapeyron effects may play a role cannot be ruled out, particularly because the glacier faces the dominant wind direction. However, we doubt whether Hardangerjøkulen stands out enough from

the surrounding topography to induce significant orographic lifting. We now mention specifically that the origin of the mass balance decrease is uncertain, and that long-term snow depth measurements on the other outlet glaciers are needed to identify the mechanism causing it.

- Last sentence: approximated by second-order polynomial vs. in caption of the figure that illustrates this (figure 2): third-order polynomial? Which one is it?
  We thank the reviewer for spotting this. Corrected to "third-order".

**Section 2.3.1:**
- p.6, l. 2: first you say that the ice cap can considered as temperate (i.e. all ice at pressure melting point) and in next sentence you mention an outlet glacier to be cold-based (i.e. ice cap is polythermal and not temperate). Not consistent. Also not very clear what has been measured and what not.
  We have reformulated this to be more precise. Midtdalsbreen may have a locally cold-based margin, but the rest of the ice cap is temperate and we think classifying the ice cap as polythermal would mislead the reader.

**Section 2.3.2:**
- Very large range for velocities for lower ablation area of Midtdalsbreen: 4-40 m a$^{-1}$ the upper part of this range is even faster than the values that you mention further for around the ELA (33 m a$^{-1}$): is this really the case? Could be due to local topography/sliding/..., but otherwise would expect higher velocities around the ELA.
  The large range in Vaksdal (2001) reflects the spatial variations in the lower ablation area. The front is very slow-moving, almost stagnant, perhaps due to the frozen bed mentioned above. The measurements from Vaksdal (2001) are summer velocities. In contrast, the 33 m a$^{-1}$ at the ELA of Midtdalsbreen include both summer and most of winter; it was measured from 14 May 2005 to 18 March 2006 (Giesen, 2009, p.47). Velocities in summer are expected to be higher than in winter, which should explain the difference. In addition, there could be interannual variations. We now clearly state the different measurement periods in the manuscript.

**Section 3.1:**
- Not fully sure about the formulation of the SIA. Typically explained more as a function of (glacier) width vs. ice thickness. What do you exactly mean by 'typical glacier length' (l.24)? How do you determine the 'characteristic horizontal scale' (l.29) for your ice cap to be 4-8 km (and the 'characteristic ice thickness to be around 200 m' (l.29)?)
  We agree with the reviewer that SIA validity is a function of the horizontal extent and ice thickness. The aspect-ratio $\varepsilon$ in Eq. (2) is a measure of this, with the underlying assumption that surface slopes are small. See also Eq. (5.5), (5.6) and (5.77) in Greve and Blatter (2009), p.63 and p.77.
    We now specify that the typical horizontal scale is based on Midtdalsbreen and Rembesdalskåka's length records from the Little Ice

Age until today (~4.5-6.5 km and ~9-11 km, respectively). The "typical" vertical scale is more challenging to quantify due to the highly variable bedrock topography and is therefore estimated qualitatively by looking at ice thicknesses around the ELA. We now also include brackets in Eq. (2), so that ε = [H]/[L], to highlight that [H] and [L] are typical values and does not represent any particular part of the glacier.

- As I indicated before, given the model you use, a comparison between SIA and HO would have been interesting (and computationally feasible)
  See previous comments on SIA/HO.

- Would recommend to also have a look at recent paper by Kirchner et al. (2016) who review in-depth the differences between models of different complexities for longer time scales. Interesting elements that you could (/should?) add when discussing the SIA / HO-FS differences (not only here, also for other parts in text)
  We are thankful to the reviewer directing us to this relevant paper, which suggests that SIA/FS differences may be larger than expected from theory and that FS may be needed in more dynamic regions (ice streams, ice shelves, areas of fast flow). We now mention this study here and in our Discussion, but as the reviewer suggest, we choose not do discuss SIA/HO/FS differences extensively since we have not performed a comparative study, as mentioned before.

**Section 3.1.1:**
- Be consistent in formulation with $\tau, \tau, \tau_d, \bar{u}_d, u_b, \bar{u}, u,$ which is not the case at this point.
  Changed.

**Section 3.1.3:**
- l.22-24: really need the lower resolution? Would expect higher resolution to be computationally feasible. If opt for low resolution, would do (one) higher-resolution run for comparison also.
  We thank the reviewer for this suggestion. We are performing experiments to test convergence on mesh resolution. Preliminary results show that the total volume varies by less than 5%; details will be given in the revised manuscript.

- l.25: need such a small time step?
  We also tried longer time steps, but numerical instabilities arose already at 0.025 years, so we settled on 0.02.

**Section 3.2.1:**
- l.29-30: repetition (+ see earlier comment: are this 44 or 45 years of measurements?)
  We choose to keep this sentence, since we do not think it is obvious from Section 2.2.3 how and what part of the available SMB data is used in our model.

It is 45 SMB years, as stated in previous response.

- SMB forcing: very simple. Not sure about applicability for other periods in time. Cannot catch many processes that are important and probably very different under other climatic conditions (changes in albedo, changes in refreezing,…etc.)
Our choice of a simple SMB profile is discussed in the Specific comments, Section 5.3, below.

- p.9, l.3-5: elaborate. Not clear at this point.
We now elaborate this further, stating that 'The averaged 35-year specific mass balance profile corresponds to an annual mass balance for Rembesdalskåka of -0.175 m w.e. We therefore shifted this profile by +0.175 m w.e. to obtain $B_{ref}$.

**Section 3.2.2:**

- Rate factor does not only depend on ice temperature. Important, but not the sole parameter. This is for instance clear from the fact that a wide range of rate factors is used for temperate glaciers, while the temperature is always at the pressure melting point. In your discussion and rationale the focus is too much on temperatures, be careful. l.21: "corresponding to ice temperatures" → "roughly corresponding to ice temperatures".
We agree that "corresponding to ice temperatures" is confusing wording, since the rate factor does not only depend on ice temperature, as also mentioned by reviewer 2. We now also state that rate factor can depend on ice fabric and impurities (and possibly other factors).

- l.30: "Based on figure 3": cannot base yourself on figure to conclude something. You base yourself on the experiments (their outcome) and the figure illustrates this.
Good point, now clarified.

**Section 3.2.3:**

- Again start with a repetition: overlap with section 2.2.2: should reorganize this to make text more consistent.
We now more clearly separate data/reconstructions (Section 2.2.2) and model forcing (Section 3.2.3).

- l. 19: "adds additional uncertainty and unnecessary complexity": be more specific. Not sure some additional complexity is unnecessary, could very well be needed to capture some processes…
We agree that complexity is not necessarily negative. We now clarify that our simple, linear SMB forcing for the Holocene is not only a result of poorly known climatic/SMB conditions in the past. It is also a deliberate strategy we choose to assess/isolate any non-linear, asynchronous behaviour in a clean way.

Section 3.2.4:
- Last sentence (l.1-2, p.11): repeat yourself again. Would remove this.
  Good suggestion, now removed.

Section 4.1:
- l.5: again a repetition.
  We thank the reviewer for highlighting this. We now remove repetitions and focus on the actual results.

- l.6-7: you "demonstrate" that growth is non-linear. Of course, this is not an idealized setting, so rather trivial that growth is non-linear. Is this really "demonstrating " something? Lines that follow: long part to say little.
  We agree that this is not appropriate wording. We "find" that the growth is non-linear.
    Only a theoretical case would be perfectly linear, so the reviewer is correct in that we expect a temporally variable response in the real case. However we do not think it is obvious that Hardangerjøkulen would grow in this stepwise manner, and even so, the timing and its relation to bed topography and the SMB-elevation feedback are interesting aspects of Hardangerjøkulen's history and have implications also for the long-term evolution of other ice caps.
    We now also improve clarity in this section by more clearly linking it to subsequent sections and Discusison.

Section 4.2.1:
- l.28: start with another repetition.
  Deleted.

- l.30-31: have a very large spread. Of course, large ensemble, most are wrong (too stiff/slow or too viscuous/fast): the range mentioned depends fully on the size of your ensemble and per se does not mean anything.
  This is a valid point. We now rather specify which range of parameter gives plausible results for the ice cap volume/extent.

Section 4.2.2:
- Very descriptive, chaotic and lacks structure. Should reorganize this and be more specific (to-the-point) to be clearer.
  We have rewritten this section focusing on clarity and now keep a chronological structure.

Section 5.1:
- l.30-31: "this is not surprising" → would reformulate this.
  Now reformulated to "This can be explained by…" We also use related advice from reviewer 2, stating that surface velocities are a function of both A and β, and the same surface velocities can be kept by a reduction of sliding and increased shear (or vice versa). However we do not

> calibrate our models against surface velocities (because of poor data coverage, as pointed out in the manuscript).

- First paragraph: discussion about (basal) velocities: have very little information (especially when it comes to basal velocities) (as you mention yourself) → discussion is not really relevant.
  It is true that little is known about (basal) velocities. We now therefore only explain the model behaviour itself, and stress that more velocity data would be needed to assess deformation/sliding in more detail.

- l.6-11: Rate factor is not only related to temperature (see earlier comment). → l.14: "corresponding to -3°C": directly relating to temperature is probably not relevant/correct.
  We agree that this was not appropriate, see response to earlier comment (Section 3.2.2).

- l.20-26: weak description. Many words to say little. In the end you say: if fast → thin / if slow/stiff → thick
  We now reduce and clarify this section significantly.

- l.30 (p.13) → l.2 (p.14): mention something interesting. Would do this here. At this point the manuscript introduces a model and a (pretty straightforward) calibration/validation (and the evolution for this specific ice cap): what is the added value of this study compared to earlier studies?
  We believe that we perform a robust calibration with the data we have available. The available (velocity) data are not sufficient to constrain the dynamic parameters to a narrower range.
  We agree with the reviewer that the implications of our study were not clear. As mentioned above, our new, dedicated subsection on transferability/applicability improves this.

Section 5.2:
- Long section about sliding: do almost not have any information. Based on your modeling → cannot really learn anything new about sliding for this ice cap. Results are simply related to your model setup and in the end your finding (which you mention further: that a lot of different combinations for your rate factor and sliding parameter are possible) is logical (as both flow and sliding have similar spatial patterns in your setup) and this was already demonstrated in earlier studies.
  We agree with the reviewer here, and have strongly shortened this section, as also suggested by reviewer 2.

- Comparison with other studies on ice sheets. Is this relevant? Totally different setting, other mechanisms for water to reach the bed (/being locally produced).
  Good point, we now focus on other ice caps and outlet glaciers.

- l.18:"It is therefore not surprising" → change

Changed to "...which probably explains why Hardangerjøkulen is more sensitive to the sliding parameter value than Langjökull."

- l.26-29: relationship sliding and geometry: from theoretical perspective. This is not a "finding" from your study..
We are not sure what the reviewer means here, studies in l.24-27 are model studies of paleo-ice sheets. We do get a thinner ice cap with increased sliding, and we find value in highlighting previous work. We have now however omitted some of the details, since the papers cited studied ice sheets and not ice caps.

- l.28: "Thus, for whatever the cause,.." → If you want to know the cause: have a look into ice flow theory.. + not kind of language expected in scientific text ("for whatever the cause"..)
We agree that this was not appropriately phrased and have deleted this formulation.

- p.15, l.3-4: indeed. A whole section to say very little..
Now shortened.

Section 5.3:
- l.6-9: repeat yourself.
Good point, now removed.

- l.10-13: SMB vs. elevation: too simple here. What about albedo, refreezing and for instance insolation (expect very different SMB vs. elevation for a surface oriented to the South and one oriented to the North...)
We appreciate that the reviewer suggests several relevant processes for the SMB. However, we deliberately chose to use a simple mass balance formulation, to focus on ice dynamical, long-term response to spatially homogeneous changes in the forcing. We justify this formulation based on results presented in Giesen (2009) and Giesen and Oerlemans (2010). They simulated the ice cap evolution through the 20th century with the simple SMB profile used here, as well as with a spatially distributed mass and energy balance model. Differences in ice volume and outlet glacier lengths at the end of these simulations are present, but small. Even when including an albedo scheme, a spatial precipitation gradient, and aspect and shading effects on insolation, the modelled lengths of Rembesdalskåka and Midtdalsbreen cannot both be matched with the observations. This suggests that this should not be attributed to the SMB, but to other factors.

  As Giesen (2009) and Giesen and Oerlemans (2010) already studied spatial variations in the SMB, our aim is not to repeat their analyses. Instead we include the results relevant for our study in this Section. Hardangerjøkulen has a gently sloping surface and is not surrounded by high mountains. Therefore, topographic effects on the insolation result in small spatial variations of the SMB are between -0.1 and +0.1 m w.e. for the vast majority of the ice cap, only two outlet glaciers oriented south show larger deviations locally. Under a realistic 21st century scenario,

Giesen and Oerlemans (2010) show that lowering the ice albedo from 0.35 to 0.20 only leads to a 5% larger volume decrease of the ice cap. Furthermore, even in a considerably warmer climate with a smaller ice cap (with continuously updated topographic effects on solar radiation), the SMB gradient with elevation was close to the present-day value. We conclude that using a SMB profile only dependent on elevation is a good approximation for Hardangerjøkulen, even in a different climate with a smaller or larger ice cap.

- l.19-20: what do you mean? Be more specific.
  We now specify that such studies would need to be coupled reconstructions of (winter) precipitation and glacier variations, on both sides of the ice cap. We leave it up to the reader to decide exact what type of proxy methods would be best suited for such reconstructions; their details can be found in the cited paper.

- l.23-24: indeed. Could this not be done?
  Further snow and SMB studies aiming to quantify the spatial accumulation variability require laborious efforts. Since the interannual variability in SMB in general and winter accumulation in particular is large (Giesen, 2009; Giesen and Oerlemans, 2010), such a campaign would have to run over several years. We now specify that with "further snow and mass balance studies" we mean field measurements.

- l.31-33: snow redistribution. Could indeed have an effect. But probably smaller effect than the large errors induced by your other approximations.
  Good point. We now make clear that here we explain the observed SMB rather than our model results, and combine this with the paragraph above.

- p.16, l.3-8: not convinced that this error is that large compared to the magnitude of errors induced by your simple modelling..
  In our opinion this error is large. However, because it only applies to the last years of our simulation period, the effect is small. We think this SMB data correction from the Norwegian Water and Energy Directorate (NVE) is worth to include.

- l.9: "works well": not sure..
  See above comments. As mentioned previously, our goal is not to reconstruct SMB for the Holocene and LIA, but to assess the long-term dynamic response to a simple climate forcing.

- l.13-17: of course, so would need albedo in model! Does not have to be a very complex model where a lot of data is needed for validation/calibration (e.g. model solving the full energy balance): this can be done in a rather simple way, but which is very effective (e.g. PDD model, T index model, simple energy balance model,…) (e.g. Braithwaite, 1995; Hock, 2003; Oerlemans, 2001)

Including any kind of albedo scheme would indeed add detail to the simulations. However, we do not aim to reconstruct/project the mass balance details of the ice cap changes. Our approach is to force the model with mass balance anomalies and not with temperature and precipitation records.

Since concern about the SMB forcing arises at several places, we have summarized the effects in our new discussion of SMB. As mentioned above, even with a full surface energy balance model (Giesen and Oerlemans, 2010), changes in the SMB vertical gradient are small, so the profile we use is probably also a good approximation for SMB in different climates. Of course, there will be effects of all the processes not included, but they will be second-order.

- l.21-23: Holocene changes in climate are strongly influenced by changes insolation, so this should be taken into account. Could be done with simple parameterization also.
  We believe that we have justified our choice not to include insolation changes in the original manuscript, l. 19-23. See also above comments on radiation.

**Section 5.4:**
- Discussion on ice dynamics, while you do not really have the material to discuss this. This is mostly a reference to the literature. A pity, given the fact that your model can be run in HO and a comparison can be made...
  As said before, we agree with the reviewer that we put too much focus on SIA/HO/FS, since we do not perform comparative tests (for reasons mentioned before). We therefore have strongly shortened this section.

- p.17, l.1-4: you discuss the effect of sliding and the deterioration of the SIA as this increases. Is indeed true. Then say that because you do not necessary have information → cannot draw conclusions. This is true, but I think that the main reason why you cannot draw conclusions is simply because you do not have a 'reference run' (a HO/FS run) to compare to.
  See previous responses on SIA/HO/FS.

- l.5-13: this is not a discussion of your results.
  Good point, we now avoid making general statements and only discuss our own results and their implications.

**Section 5.5:**
- l.23: effect SIA. Of course true, but the effect of SIA/HO-FS is very limited compared to other errors and uncertainties. Over Holocene timescale the SMB (where uncertainties are large) will have much larger effect than dynamics on the evolution/growth.
  We agree that SMB is more important than SIA/HO/FS on Holocene time scales, which we mentioned before. We now state this clearly in the text.

**Section 5.6:**

- • l.4-9: growth → very descriptive and site specific. What is added value for reader?
  We agree that this was unclear. We now discuss the non-linear, asynchronous growth in our new transferability/implications subsection. We also analyze the volume-area variations (Fig. 10) in more detail and in light of volume-area scaling relations in the literature.

- l.21-22: "this asymmetry illustrates that proxy records representing different parts of an ice cap may lead to substantially different conclusions about ice cap size through time" → of course. Rather trivial.
  While intuitive, we do not think this is appreciated in the literature of glacier reconstructions, where conclusions about past glacier activity and climate are sometimes drawn from a single outlet glacier of a larger ice mass.
  We realize that there is not much input from any of the reviewers on the proxy/paleoglaciological relevance of the study, for example no comments on the opposite asymmetry during growth and retreat. Since we think these aspects are important, we should have emphasized them more and thus now make the long-term Holocene evolution and the effect of SMB-elevation feedback more visible in the Results and the Discussion.

- l.23-32: long passage with little information.
  l.23-28 contains some in our view relevant previous studies worth mentioning, and we think our findings about overdeepenings and glacier advance complements/build on these.
  We agree that l.29-32 was vague and is now shortened and more to-the-point.

- p.19, l.1-27: many words about response time to in the end say very little. Do not have experiments to elaborate on this. Could spend a few words on this, but not whole section.
  We agree with the reviewer, and now keep it short and specific with regards to our results.

- l.28 → p.20, l.1-2: not sure that your results support this. Rather speculative.
  We have not studied erosion, sediment transport and deposition in our study, so we agree with the reviewer that we should be careful about drawing specific conclusions on sediment-based reconstructions. Still, we think our out-of-phase evolution of volume and area for many centuries (l. 22-23; Fig. 10) suggests that linear assumptions between basin size (area), ice volume (mass balance), climate, and their proxies should be challenged.
  We are now more specific on this and less speculative when it comes to sedimentation.

**Section 5.7:**

- l.7: effect proglacial lake. Can have an effect, but expect this again to be much smaller than other model uncertainties.
  We agree, and now point this out.

- l.14: "in our view a step forward": not sure. Even if would be the case, you should maybe not write this down and let the reader decide for himself whether he thinks this is new/novel/better than methodology applied in other studies. First focus should be a carefully calibrated/validated and robust setup, supported by field data, and not sure whether this is the case in this study.
  We agree with the reviewer that this was not appropriate wording. We think however that our methodology, results and their implications have value for other studies, which we also highlight in our new transferability/implications subsection mentioned above.

**Section 5.8:**

- l.25: you "show" that ice cap is very sensitive to change in climatic conditions. Trivial: of course, it is an ice cap. Importance SMB-elevation feedback. Has been analyzed in (far greater) depth and from theoretical point in the past. Have a look at some of the 'classic' papers on this (Lee and North, 1995; Mahaffy, 1976; North, 1984).
  We now analyse the SMB-elevation feedback in more detail and have moved it to the Results. The sensitivity to SMB is exceptionally strong for Hardangerjøkulen, the feedback is crucial to this sensitivity. See previous comments on SMB-elevation feedback.

- p.21, l.1-4: again rather trivial. What's new about this finding?
  Perhaps it is trivial that the relation is linear without including the feedback, but we mainly use this experiment to illustrate that it is indeed the feedback that makes the ice cap so sensitive. To illustrate this difference, we now show the modelled transient ice volume evolution in response to SMB perturbations of the present-day ice cap, without the SMB-elevation feedback, in Fig. 12. We have also moved it to the Results section.

- l.9: 750 years to disappear. Too precise. Would change this to "around 750 years"
  Changed as suggested.

- "As evident from Collins et al. (2013), we expect a warming scenario": strange formulation.
  We agree, and have changed this to "Future projections suggest a warming scenario for southern Norway"

- l.11-21: do not really discuss your own results, not based on your simulations.
  To keep the focus on our own results, we now exclude most of the paragraph about the future.

- l.22-28: what's new?
  Now specified that our study provides new detail on the transient evolution/growth and retreat during Holocene and its relation to bed topography/SMB-elevation feedback. We now also consider the reconstructed disappearance into perspective of mass balance sensitivity. We choose to keep the line about future warming and refer to Giesen and Oerlemans (2010) for future projections.

**Conclusions:**

- Start from ice-free in Holocene: do you also get this if would start simulations earlier and force with a palaeoclimatic record? Would be an interesting experiment..
  This would indeed be interesting, but it is not within the scope of our study. We see this as a suggestion for a future study, where a (simplified) mass and energy balance model is used to study the full Holocene evolution of Hardangerjøkulen, forced with paleoclimatic records of temperature, precipitation and insolation. However, a challenge would be what ice cap state to start with, since no good estimates on ice volume/extent of Hardangerjøkulen exist prior to the mid-Holocene ice-free period. In this study, we start from ice-free conditions, because this is the most robust route to study the Holocene ice cap evolution from 4000 BP onwards.

- p.22, l.3-6: this is not something new. Not a finding from this study.
  We agree with the reviewer and now deemphasize this. Still we would like to encourage other studies to keep calibration ensembles during transient simulations, so we have kept this in our conclusion.

- l. 9-14: SMB-elevation feedback exists for ice cap. You show this, but do not really add anything new to the theory related to this.
  These lines do not specifically refer to the SMB-feedback, and we think what is mentioned is relevant for other studies of ice caps, as mentioned before.

- l.15-17 + l. 24-26: site specific → what is the more general interest?
  This is indeed not clear. We have completely reworked the Conclusion aiming to be clearer about our findings and what the value/implications are.

- l.27-31: strange way to end your conclusion..
  We agree and have integrated this into the Conclusion.

**Figures:**
• Nice and clear figures in general.
We thank the reviewer for this. We have made some improvements anyway:
- Fig. 2: swapped red/blue colours in legend, as they did not correspond to the lines in the figure.

- Fig. 6: plotted individual simulations to indicate the distribution. Used different colors for different rate factors, same color for different sliding parameters within the same rate factor.
- Fig. 10: Added a related figure showing the relationship between volume and area, with relevance for volume-area scaling methods
- Fig. 12: included simulations excluding the SMB-elevation feedback, to add detail to this feedback in the manuscript, as suggested by the reviewer.

**References**

Braithwaite, R. J.: Positive degree-day factors for ablation on the Greenland ice sheet studied by energy-balance modelling, J. Glaciol., 41(137), 133–160, 1995.

Hock, R.: Temperature index melt modelling in mountain areas, J. Hydrol., 282(1-4), 104–115, doi:10.1016/S0022-1694(03)00257-9, 2003.

Huss, M. and Hock, R.: A new model for global glacier change and sea-level rise, Front. Earth Sci., 3(September), 1–22, doi:10.3389/feart.2015.00054, 2015.

Jouvet, G., Huss, M., Blatter, H., Picasso, M. and Rappaz, J.: Numerical simulation of Rhonegletscher from 1874 to 2100, J. Comput. Phys., 228(17), 6426–6439, doi:10.1016/j.jcp.2009.05.033, 2009.

Kirchner, N., Ahlkrona, J., Gowan, E. J., Lötstedt, P., Lea, J. M., Noormets, R., von Sydow, L., Dowdeswell, J. A. and Benham, T.: Shallow ice approximation, second order shallow ice approximation, and full Stokes models: A discussion of their roles in palaeo-ice sheet modelling and development, Quat. Sci. Rev., 135, 103–114, doi:10.1016/j.quascirev.2016.01.013, 2016.

Lee, W. and North, G.: Small ice cap instability in the presence of fluctuations, Clim. Dyn., 1995.

Leysinger Vieli, G. J.-M. C. and Gudmundsson, G. H.: On estimating length fluctuations of glaciers caused by changes in climatic forcing, J. Geophys. Res., 109(F1), F01007, doi:10.1029/2003JF000027, 2004.

Mahaffy, M. W.: A three-dimensional numerical model of ice sheets: Tests on the Barnes Ice Cap, Northwest Territories, J. Geophys. Res., 81(6), 1059–1066, doi:10.1029/JC081i006p01059, 1976.

North, G. R.: The Small Ice Cap Instability in Diffusive Climate Models, J. Atmos. Sci., 41(23), 3390–−3395, doi:10.1175/1520-0469, 1984.

Oerlemans, J.: Glaciers and Climate Change, Balkema., 2001.

Raper, S. and Braithwaite, R.: Glacier volume response time and its links to climate and topography based on a conceptual model of glacier hypsometry, Cryosph., 183–194, 2009.

Zekollari, H. and Huybrechts, P.: On the climate–geometry imbalance, response time and volume–area scaling of an alpine glacier: insights from a 3-D flow model applied to Vadret da Morteratsch, Switzerland, Ann. Glaciol., 56(70), 51–62, doi:10.3189/2015AoG70A921, 2015.

Ziemen, F., Hock, R., Aschwanden, A., Khroulev, C., Kienholz, C., Melkonian, A. K. and Zhang, J.: Modeling the evolution of the Juneau Icefield between 1971 and 2100 using the Parallel Ice Sheet Model (PISM), J. Glaciol., 2016.

---

## Author Comment (AC2) · 19 Aug 2016

**Response to Reviewer #2**

We would like to thank the referee for insightful and constructive comments. Several valid points are raised, which we respond to in a point-by-point manner below (blue). We have also made substantial efforts to rework the manuscript to improve structure and clarity.

On behalf of the authors,
Henning Åkesson

**General Statement**
The paper presents some modelling results, which concern the growth and the retreat of the Hardangerjokulen ice cap, South Norway, from the mid-Holocene to the presentday. To do so, the authors have used the Ice Sheet System Model to simulate the dynamical evolution of the ice cap. The model accounts for internal ice dynamics (SIA), linear basal sliding and surface mass balance. Dynamical parameters (sliding and shearing of ice) are first calibrated at present-day, and secondly in transient runs. The transient simulations indicate an asymmetry between southwest and northeast section during both advancing and retreating stages. This study is in line with number of previous papers (referenced in the manuscript), which present modelling results for a specific glacial area, and compare the results to field evidence. I have no doubt that the methodology can be successfully applied to gain insights about the chronology of the advances and retreat of this particular ice cap and complement the geormophological information already available. Unfortunately, I find the present manuscript hard to follow so that the main achievement of the paper is somehow hidden. One reason to explain that is: the paper does not follow any clear continuous line, which should bring the reader from the original investigated problem to some final conclusions.

We agree with the reviewer that the original manuscript could have been clearer and more concise. We have now reworked the Abstract, and completely rewritten the Introduction, clearly stating the scientific questions we address and the reasons for doing so. We have shortened some sections in the Methods, as suggested by the reviewer (specifically on SIA, see below). In the Discussion, subsections are now more explicitly linked together and paragraphs not following the main aims and scope of the paper have been deleted. We have also added a dedicated subsection on transferability/implications in the Discussion. In the Conclusion, we highlight our main findings more directly linked to the scientific aims outlined in the Introduction.

The paper spends a lot of sentences to discuss things of little importance/originality or already debated many times, and this strongly harms the overall reading. Unfortunately, the most interesting results arrive at the end of the paper, so that it is likely that most of readers won't reach this point (being discouraged by too many unnecessary discussions).

We thank the reviewer for highlighting this. We hope that our reorganization of the manuscript will guide readers to these also in our view important results on hysteresis and high mass balance sensitivity.

In addition, the paper shows number of inaccurate/inappropriate/awkward sentences, which harm the overall argumentation (see some examples below). I believe that the paper must be rewritten before to be reconsidered for publication. This including a substantial shortening (removing unnecessary/distracting parts, and better emphasizing the main outcomes). I hope that my next suggestions will help the authors to achieve this task.
We agree with the reviewer that these sentences arise from unclear writing. We reply specifically to the examples given by the reviewer below, and have kept clarity in mind when revising the rest of the manuscript.

**Major concerns:**
- Section 5.2 is a typical example of section, which really slow down the reading because of a lack of originality and importance for the present study. I would recommend to remove it, or to keep the most relevant information (maybe to be merged with Section 5.1). I believe that Section 5.3 could be more efficiently and more concisely rewritten. More generally, the whole Section 5 should be "optimized"
We are thankful to the reviewer for these concrete suggestions. Section 5.2 is now greatly shortened with relevance to the present study in mind, and merged with the Discussion of the dynamical parameters in Section 5.1. Further, Sections 5.3 and 5.8 in the original manuscript both discussed mass balance. They are now rewritten and combined into one section.  We also combine Section 5.5 and 5.7, because they both discuss the impact of the linear build-up phase.

- The manuscript contains hazardous/inaccurate statements, and sometimes awkward/dangerous assessments. However, I believe this is more unfortunate formulations rather than misunderstandings by the authors. For instance:
  - p. 7: "Where bed and surface topography is complex, lateral drag and longitudinal stress gradients may become important. Still, the SIA has proven accurate in representing glacier length and volume fluctuations on decadal and longer time scales." is more confusing (and even contradictory) than useful.
    We agree with the reviewer, and now shortly justify SIA in light of Hardangerjøkulen's characteristics, our time scale of interest and the references given.

  - – l.23 p. 8: "Even though our surface digital elevation model (DEM) has higher resolution than this (100 m), we choose the highest mesh resolution to be 200 m, since this is more in line with the assumption of the SIA" In what the mesh size and the physical model (here SIA) are connected?
    We thank the reviewer for highlighting this. The stress balance of SIA is completely local. Using a very high resolution for SIA increases the risk of unphysical stress gradients/velocities due to local variations in bed topography. We avoid this by smoothing the DEM. As we already point out, the rationale behind the lower mesh resolution is also to save

computational resources. As mentioned in response to reviewer 1, we now also carry out an analysis on mesh convergence. Preliminary results show that volume varies by less than 5%; details will be given in the revised manuscript.

– – "SIA is viable to use if interests are climatic rather than ice dynamics." should be more accurate.
See above comment on SIA.

– – "By investigating a small valley glacier in the Canadian Rocky Mountains and neglecting basal sliding, Adhikari and Marshall (2013) suggested that SIA performs well in less 'dynamic' settings, while the results compared to HO/FS diverge for more 'dynamic' situations." is a striking example: of course, if there is less dynamics, then the errors related to the dynamics gets less visible!
This is a good point and we now refrain from make such general statements. As requested by reviewer 1, the Discussion section on ice dynamics is shortened, since we do not have much available data to constrain our results to, and we do not run comparative tests for SIA/HO/FS due to reasons given below.

– – l.1 p.17: "It is challenging to assess how much sliding there could be before SIA validity deteriorates, but it likely depends on the climatic and glaciological setting. " is inaccurate
We agree that this statement is rather confusing and is now removed. We now deemphasize our discussion on sliding, since we do not have the available data for validation of basal motion.

– – l. 23 p. 17 "we are aware of the limitation ... Therefore, the actual rate of advance may differ...". The actual rate of advance may differ because you are aware of the limitation of the SIA?
This is indeed unclear wording. We now change this to "...the actual rate of advance may differ ..., because SIA has limitations in the steep terrain..."

• The paper is poorly structured. Some information come repetitively in the paper (as the justification of using the SIA). The discussion (Section 5) looks more like a list of items without connections between the subsections. The last paragraphs of each subsection of Section 5 state some recommendations for future studies. I think this is not the right place, such statements should rather appear in a dedicated "perspective" closing section.
Substantial efforts have gone into restructuring the manuscript; see comments above. Regarding "future work", we integrate these more appropriately into the text. However, we do not think a dedicated "Perspective" section is necessary.

• The dynamical model is essentially based on the most simple existing model, namely the Shallow Ice Approximation. Even if this is a surprising choice (regarding to the capabilities of ISSM, and other higher order models freely

available nowadays), I find unnecessary to describe in details the well known SIA so that Section 3.1 can be strongly shortened. In addition, there are several clumsy attempts to justify the use of the SIA throughout the whole paper. The uncertainty due to mechanical simplifications cannot be quantified since no comparative tests are done with higher order solutions. As a consequence, I don't see the point of discussing so much in details this assumption, while a simple referencing to previous comparative studies (e.g. Lemeur and al) would be enough.

We agree with the reviewer that the details of the SIA theory can be omitted. However, we do believe that there is a value in justifying our choice of SIA, especially since both reviewers suggest that a HO/FS model should be used if available. As mentioned in our response to reviewer 1, the SIA is considerably cheaper and allows for ensemble and longer time scale studies. We believe that HO/FS is unnecessary for the long time scales studied here, on this ice cap lacking areas of fast flow. This is in line with what previous studies have shown (referenced in the manuscript). Even if we had attempted a SIA/HO comparison within ISSM, it would not be straightforward. The problem is that the parameterization of the basal friction is different for SIA and HO in ISSM; SIA parameterizes basal velocities and HO parameterizes basal stress. We therefore do not think a SIA/HO twin simulation would be informative.

- All what concerns the calibration to ice flow and sliding parameters should be strongly shortened since this problem has been presented many times so that only the result matters. Also, I am not really convinced by the "best-fit" pair of parameters, which is chosen among all those which minimize equally the RMSE. If I understand correctly, the 'best-fit' parameters are chosen according to the temperature of the equivalent rate (Arrhenius) factor A, which should correspond to temperate ice. This is a very weak argument, which cannot be used to constrain A. Most of ice flow models are tuned through enhancement factors, this indicates that one cannot rely directly on the exponential formula for A(T) given in (Cuffey and Paterson, 2010, p. 73). Say differently, the formula works in a relative way (after tunning), but not in a absolute way.

The discussion on the dynamical parameter calibration can certainly be shortened. The inability to find a unique parameter combination is important but indeed not new. However, many studies do not keep their parameter ensemble during transient runs like we do, and risk loosing some important information on parameter sensitivity, so we still believe that our approach requires some attention.

  We completely agree that the Cuffey and Paterson A(T) formula can only be used after tuning, not in an absolute way. We also agree with the underlying statement that assuming an A according to the table value corresponding to temperate ice is a weak argument. To be clear however, we do not assume temperate ice. Instead, we tune A without any *a priori* assumption about ice temperature and use the RMSE for ice thickness as a constraint. We arbitrarily pick an A (which corresponds to T=-1 C) in the middle of the region of similar RMSE's (dark blue region in Fig. 3). Differences in RMSE within this region are within 1 m, further underlining the motivation behind

keeping our ensemble after the calibration. A comparison with a map of ice velocities (which is not available for this ice cap) would more strongly constrain A, and we try to convey this in the paper by stating that several parameter combinations give similar RMSEs for ice thickness. A key result of the paper is also that the impact of A on ice volume is large during our transient simulation over several centuries (Fig. 6), while relatively small at calibration (Fig. 3). This disparity suggests that small differences in model rheology at calibration propagate with time. This time-dependency has implications for other model studies of long-term dynamics of glaciers and ice caps.

- I don't see in what the spatial asymmetry is intriguing or unexpected since you mention several times that precipitation are asymmetric (west-east precipitation gradient).
  We should have been clearer on this. In *reality,* there is likely a W-E precipitation gradient, due to the prevailing SW-W wind direction. In the *model,* there is no such gradient. Instead, SMB is prescribed as a function only of elevation. In our case this neglected horizontal SMB gradient is an advantage, since we know that the spatial differences during growth and retreat we find by definition cannot be due to an asymmetric SMB forcing. This is why we consider our found asynchronous growth and retreat an interesting and perhaps unexpected finding.

- Figure 11 and 12 are to me the most interesting results of the paper, so that they deserve to be better highlighted. However, the authors should clarify in what these results are different from the Figures 6.7 and 6.8 of (Giesen, 2009, PhD dissertation), which is already based on the SIA.
  We thank the reviewer for these encouraging words. We now make these results more visible by moving them to the Results, and link them to the aims of the rewritten Introduction. Giesen (2009) indeed used SIA, although with a different sliding and ice deformation formulation, as well as different numerical methods (finite difference and not finite element model). Figs. 6.7 and 6.8 from Giesen (2009) have not been published in a peer-reviewed journal. Since the results in our present study support Giesen (2009), but are not identical and derived from a different model, we think these findings are worth highlighting. What is new in the present study is also the inclusion of our parameter ensemble, providing an estimate on the effect of parameter uncertainty on the relationship between SMB anomalies and steady-state ice volumes.

**Specific comments:**
- Abstract and later: I don't understand why you say that your model is 2D. The SIA provides a 3D velocity field. To me, your model is 3D.
  In a 3D model, velocities are calculated explicitly for the x-, y- and z-directions, which is not the case here. We use vertically-averaged

horizontal velocities, thus we do not resolve vertical variations in horizontal velocities. Therefore we view the model as being 2D.

- Abstract and later: You often emphasise the capabilities of ISSM to perform mesh refinements, but you never say what does it brings to the study. If this is not relevant, it should be in the abstract.
  Good point. It adds better accuracy (200 m) around the LIA margins due to the mesh refinement there, but when the glacier is smaller/larger the accuracy is reduced (400-500 m). This is now pointed out in Methods.

- Eq. (1): I don't think it is necessary to repeat the formula used to reconstruct the ice thickness (or equivalently the bed). Referencing would be enough.
  Referenced only.

- l. 12 p.4: I don't think that the acronym NVE was defined so far.
  Written out.

- l. 4 p.5: "At c. 4000 BP" and many other places in the text: What "c." stands for?
  It stands for circa/about/around. We now write this out the first time.

- Eq. (4) dot is missing at the end. Punctuation (coma and dots) is sometimes missing in your equations.
  Fixed.

- l.25 p.7: "SIA" must be "The SIA".
  Changed.

- l.7 p.8 "ISSM has capacities ..." this is useless information since you don't use this capability.
  We think mentioning this motivates/justifies our approach to basal sliding. We have rewritten this to: 'While ISSM has the capacity to ..., this method could not applied to Hardangerjøkulen because of the limited velocity data coverage'

- Eq. (7) u should be \bar{u}.
  Changed.

- l. 13 p. 8: is the unit of M not m a$^{-1}$? mass balance rate should rather be annual mass balance?
  Indeed, changed.

- l. 31 p. 12: "both depend on driving stress", ok but I think one can explain much more easily why several pairs of A and \beta gives similar RMSE: one can reduce sliding and increase shear of ice while keeping same surface velocities.
  Rephrased in a more concise way as suggested.

- l. 10 p. 13. As I said before, "We therefore exclude ....". I don't think you can use this argument to eliminate pairs of parameters. But instead, it sounds more reasonable to keep going with several pairs (A, \beta) which would be a set of "bestfit" parameters.
  Fair point. We now show individual runs with focus on rate factors in Fig. 6, using the rate factors from Cuffey and Paterson's table. We agree that we cannot use the argument mentioned by the reviewer and *a priori* exclude A(T=-5) rate factors. However, because we see that the smaller A(T=-3) rate factors deviate significantly from the observed ice volumes, and A(T-5) deviate even more, we choose only to show a smaller range of simulations in Fig. 6.

- l. 20 p.13 : You havn't defined $T_{ice}$.
  We now define $T_{ice}$ in Methods (l. 15 p.7 in original manuscript)

- l. 20 - l.27 p.13 : This paragraph is especially laborious to read, and can be certainly shortened.
  We have shortened and clarified this paragraph.

- l. 29 p. 13: It makes no sense to refer to numerical objects (mesh node) in this Section.
  Changed.

- l. 33 p.13: "By imposing ... ice masses". This sentence doesn't bring anything.
  We agree, and now omit this.

- l. 34 p. 13: "cold- to warm-based" I guess you refer to basal condition? If yes, you should formulate that more clearly.
  Indeed, now specified.

- l. 17-18 p. 17: ", since the ice present is divided ..." I don't understand the meaning.
  Changed to "split up into several separate glaciers"

- You should maybe rename Section 5.3 or 5.8, since it seems (from the names) that they address the same issue.
  These sections are now shortened and merged.

- l. 1-4 p.21: why don't you show the results?
  This is a good point. We have added simulations without the SMB-elevation feedback in Fig. 12. One of our main conclusions is that Hardangerjøkulen is so sensitive because of this feedback, and we now also illustrate this in the new Fig. 12 as suggested by the reviewer. We have also moved it to the Results to increase visibility.

---

## Author Comment (AC3) · 19 Aug 2016

**Response to Short Comment #1**

We would like to thank Dr. Luening for his Interactive Comment, and respond to his suggestion below in blue.

On behalf of the authors,
Henning Åkesson

While studying this manuscript I could not find any hint towards the Medieval Climate Anomaly (MCA). This is a phase which was anomalously warm in the study region and which was generally associated with a major glacier melt episode. Why is the MCA not featured in the article? The most likely driver of the MCA warming is high solar activity during this phase, requiring an adequate radiative forcing to produce an effect in the theoretical models. I strongly recommend to include a discussion on the MCA in the paper. Reference to the Little Ice Age requires an analysis of the preceding climatic event, i.e. the MCA, otherwise the discussion is incomplete.

We thank Dr. Leuning for raising this concern. The Medieval Climate Anomaly (sometimes referred to as the Medieval Warm Period (MWP) in the literature) is indeed an important climatic period in the recent past which, as Dr. Leuning points out, is associated with glacier retreat.
   Nevertheless, there is little information to validate Hardangerjøkulen's response to a climate/mass balance forcing during this period. The same reasoning applies to the earlier Holocene, as mentioned in our response to the two reviewers. We do not intend to add mass balance variations/variability without constraints, and we also wish to keep our mass balance forcing simple in order to isolate the effects of bed topography and the SMB-elevation feedback. We also do not consider assessment of the responsible climatic forcing(s) behind the Little Ice Age, its preceding or following climates as a central aim of our study. We therefore do not think discussing the MCA/MWP is relevant for our purposes.

---

## Author Response (AR2)

**Very minor comments/corrections by editor**
-p. 4 line 29: 'experienced' (not 'experieneced')
Changed.
-p. 11, line 16: 'faster sliding': a bit awkward, 'enhanced sliding' maybe better.
Changed.
-p. 14 line 29: should be 'were' not able … rather than 'was' as there are two authors.
Changed.
-p. 19 line 3: again should be '…assumptions do' rather than ''…assumptions does'
Changed.
-please acknowledge the reviewers in the acknowledgements
Reviewers and the Editor are now acknowledged.

**Response to Reviewer #1**

We thank reviewer 1, Andy Aschwanden, for constructive comments and appreciate recommendations for relevant previous studies to compare our findings with.

We reply to comments below in a point-by-point manner.

On behalf of the authors,
Henning Åkesson

The manuscript is generally well written, flows nicely and is easy to follow and now has a clear(er) focus. The authors have carefully addressed the reviewers' comments. While the reviewer's comment "This setup, with a focus on the long-term evolution of the ice cap, is interesting to get an insight in the dynamics of this ice cap and the important role of the surface mass balance (SMB) and its feedback with elevation. However, the authors do not really dig into these concepts" still holds true, though to a lesser degree, the authors explicitly state their focus on long term reconstruction.

My main comment is that the manuscript could be made more relevant and broadly appealing by putting the results in a wider context and contrasting/comparing their findings with other recent work on ice caps and icefields. For example, recent publications assess the stability of the Juneau Icefield (Ziemen et al, 2016) and Yakutat Glacier (Truessel et al, 2015). Both glaciers are in south-east Alaska in a similar climate setting, but Yakutat Glacier is expected to disappear by the end of the 21st century while the Juneau Icefield is expected to survive. Ziemen et al (2016) have done many modeling experiments that are quite similar to this study (e.g. sensitivity to sliding, regrowth capability under different climate scenarios). I find it very interesting that for present-day climate, Hardangerjøkulen ice cap only grows to 20% to its current volume when starting from ice free conditions, while the Juneau Icefield's regrowth volumes are close to the steady-state volumes when starting from present-day (Fig 11 in Ziemen). Other similarities include the sparseness of the ice thickness data. Ziemen showed, using a very crude sensitivity experiment, that uncertainties in ice thickness have a strong impact on the simulated volume evolution. Very recently, Gilbert et al (2016) studied the evolution of the Barnes Ice Cap on Baffin Island, corroborating some of the findings of Mahaffy's seminal 1976 paper, and concluded that the ice cap will disappear within this millennium. I hope this will help expand on the concluding statement (P20,L14) "We expect that ice caps with comparable geometry elsewhere may display similar sensitivity and hysteresis".

We appreciate the reviewer's directions to several recent relevant papers. In the revised manuscript we discuss our work in light of these recent studies on ice caps and ice fields (see Section 5.3 in the Discussion).  We have changed the concluding statement to "Our experiments suggest that the present-day ice cap is in a mass balance regime where it will not regrow once it has disappeared. We thus find that Hardangerjøkulen displays strong hysteresis and that the

interaction between hypsometry and mass balance-altitude feedback controls its behavior."

Figure 2, SMB-altitude profile and P15, L19-20: I'm not quite able to figure out how the profile was derived from observations without reading all the underlying literature, so this might be a red herring: looking at the geographical setting, I can imagine that the ice cap experiences orographic precipitation at least to some degree, where the windward side receives higher precipitation than the lee side. If more high-altitude area is in the lee side, averaging could lead to a decrease in SMB at high altitude. Schuler et al (2008) studied orographic precipitation effects on the Svartisen ice cap further north using the Smith and Barstad (2004) linear theory of orographic precipitation model.

We agree that the explanation of the derivation of the SMB-altitude profile should have been clearer, and we have now improved this paragraph.
  SMB is measured on the windward side, at the outlet glacier Rembesdalskåka, so the decrease in SMB at high altitudes pointed out by the reviewer is certainly real. Note however, that for Hardangerjøkulen most of the high-altitude area is on the windward rather than the leeward side (see Fig. 1 in the manuscript), therefore we are not sure whether the bias suggested by the reviewer due to spatial averaging and leeward effects will play a major role. As noted in the discussion (p.14, l.30-33), Giesen and Oerlemans (2010) included spatial mass balance gradients but were still not able to match the two outlet glaciers with modern observations, suggesting that this mismatch does not arise due to the mass balance forcing alone.

A side note: The authors' statement in the response letter "In addition, we think that the dependency of initial conditions for ice caps (hysteresis), illustrated in Fig. 11, has not received much attention in the literature and is relevant for modeling and reconstructing paleo-ice caps and predicting future ice cap evolution." may be true for small(er) glaciers, but not for ice sheets. We've been pushing this idea for more than 5 years now; see Aschwanden, Aðalgeirsdóttir, and Khroulev (2013) and Aðalgeirsdóttir et al (2014).

We thank the reviewer for highlighting these studies. We certainly acknowledge that prognostic simulations are highly dependent on initial states and that this is an area of active research. We now emphasize the case for small ice caps by adding the following at the end of Section 5.3:
"Hardangerjøkulen's strong hysteresis highlights the importance of accurately representing the initial state in transient simulations of small ice caps, as previously suggested for ice sheets, as previously suggested for ice sheets (e.g. Aschwanden et al., 2013; Aðalgeirsdóttir et al., 2014)."

**Detailed comments:**

P1, L13-14: "... that the surface mass balance-altitude feedback and ice cap hypsometry are essential to this sensitivity." The surface mass balance-altitude feedback and the glacier hypsometry are not independent of each other (or am I missing something?), the hypsometry of a glacier determines how strong the surface mass balance-altitude feedback is. If memory serves me well, this was beautifully illustrated by Rivera and Cassa (1999)

We agree with the reviewer that the hypsometry will influence the strength of the surface mass balance-altitude feedback. We now more clearly convey this by rephrasing to "the effect of the ice cap hypsometry on the mass balance-altitude feedback is essential to this sensitivity". We thank the reviewer for pointing us to the relevant paper by Rivera and Casassa (1999), which we now cite in Section 5.3 in the discussion.

P2, L3: their sea level equivalents (e.g. Grinsted, 2013; Bahr et al., 2015). It's a good practice to also cite the first manuscript that introduces a new concept (here V-A scaling).

We now cite Bahr et al. (1997) as well.

P3, L7: ...ranges form 1020 to 1865 m a.s.l....

Changed.

P6, L21: ...on a scaling analysis of the Stokes equations... ("full Stokes" still makes me cringe)

Changed.

P7, L15-20: I do not want to go off on a tangent, but why do you acknowledge that SIA is only valid for no-slip conditions and then go on and combine it with Weertman sliding? See Appendix in Bueler and Brown (2009) why this should be avoided. I do not expect this to influence your general conclusions though (in fact, you will most likely get very similar sensitivities when strictly using no-slip). Along the same lines, I agree with the discussion in the previous reviews that using a more complex stress balance is not needed here.

While stress balance simplifications (the SIA) have been addressed in previous responses to reviewers, as the reviewer points out, we appreciate that the reviewer raises the question regarding Weertman sliding and SIA. The discussion in Appendix B of Bueler and Brown (2009) concerns models where temperature varies along the glacier base. The argument is that the "switch" from no-slip to sliding conditions at the base induces "jump discontinuities" in horizontal velocities, due to the incompressibility of ice. We acknowledge that this may be an issue for polythermal ice, but Hardangerjøkulen has a base at the pressure-melting point and we therefore model ice as isothermally temperate.

P7, L23-24: "In this study, the basal sliding parameter $\beta$ is assumed spatially and

temporally constant. In reality, sliding likely varies in space and time according aforementioned factors." This two sentences do not make sense the way they are written, one gets the impression that using a constant β leads to a constant in time and space sliding, which is not true. I think you can just delete the second sentence.

We agree with the reviewer and have deleted the second sentence.

P7, L25-27: There is a much stronger argument against deriving the basal friction parameter from surface properties. Inversions for β are only valid at or around the time stamp of the data sets used for inversion. Consequently, β fields are snap shots and should not used for prognostic simulations at all, but this would be all the more severe for the long time scales (millennia) considered in this study.

We have added "More fundamentally, inverted friction fields may become inaccurate on the long time scales considered in this study."

P8, L22: Define your steady-state. See Ziemen et al (2016) near Eq. 4 why this is important to correctly interpret your results.

The reviewer here raises a valid point in that mass balance trends in different parts of an ice cap may cancel each other out, even though total volume change may suggest a steady-state. While we did not use a criterion as suggested by Ziemen et al. (2016), we ran our ensemble calibration for 2000 years for each ensemble member (cf. 1500 years in Ziemen et al., 2016), which is sufficient to ensure steady-state.

P10, L2: ...sensitivity to the choice of...

Added "the".

P12, L19,22: should be Figure 11a,b not 10

Changed numbering.

P12, 31: Our Holocene simulations shows...

Changed.

P19, L2-5: This is a rather broad brush over a large field. At least provide references (there is a wealth of literature on simple, analytical models by glaciologists including, but not limited, to: J. Oerlemans, G. Roe, W. Harrison, M. Luethi).

We agree with the reviewer. We were mainly concerned with glacier reconstructions using proxy data, but the reviewer makes a good point by including these simple, analytical models. We have rewritten this paragraph

emphasizing glacier reconstructions from proxies, as well as acknowledging previous literature as highlighted by the reviewer.

**Response to Reviewer #2**

We wish to thank reviewer 2 of the revised manuscript, Nicholas R. Golledge, for insightful and constructive comments.

We appreciate the reviewer's comment regarding the use of the SIA. As we have addressed this issue extensively in previous responses to reviewers of the original manuscript we will not repeat this discussion here. We also note that both reviewers of the revised manuscript do not view inclusion of higher order stresses critical for our application. Thus we have kept the discussion concerning the SIA in the revised manuscript as is.

We thank the reviewer for suggesting the change of tense to emphasize our aim of investigating the style and dynamics of ice cap growth through the late Holocene, rather than a detailed reconstruction of the ice cap history. We have changed this accordingly, as indicated by the examples below:

- p. 1, l. 5: " Under a linear climate forcing, we find that Hardangerjøkulen grew" -> "Under a linear climate forcing in our simulations, we find that Hardangerjøkulen **grows**"
- p.17, l.27: "we consider our continuous model **reconstruction** to be a good first estimate of how Hardangerjøkulen **grew** -> "we consider our continuous model **simulation** to be a good first estimate of Hardangerjøkulen**'s growth** from nothing to its maximum extent during the LIA."
- p.19, l.26: "Our simulations suggest that Hardangerjøkulen evolved" -> "In our simulations, Hardangerjøkulen **evolves**"
- p.19, l.27: "involved" -> "**involves**"

On behalf of the authors,
Henning Åkesson

Having read the initial reviews and the authors responses to them, I can see that the authors have made a concerted effort to simplify, clarify, and improve the paper. The major shortcoming of the study appears to be that the model implementation (using just the SIA) may produce results that are unrealistic. I think this concern still persists, but in my view the study is sufficiently conceptual that the results can be considered simply in terms of anomalies - that is, the relative changes are of interest, even if the absolute geometries and dynamics are open to question. To help convey this, one small but important modification might be to change the tense used throughout the paper when discussing the results. Rather than stating emphatically (for example), "...Hardangerjøkulen GREW from ice-free conditions in the mid-Holocene.." (L5) it might be preferable to use the present tense, and couch the statement in terms of the model results, e.g. "in our simulations Hardangerjøkulen GROWS from ice-free conditions in the mid-Holocene..:. This change of tense throughout the manuscript would make it clear that the ideas being put forward are based solely

on the experimental results, and that the authors aren't making direct claims about the 'real' ice cap. Other than this suggestion I can find little else to comment on that the two original reviewers haven't already picked up. I agree with them that the findings concerning SMB feedbacks and hysteresis etc aren't particularly novel, but they are interesting nonetheless and in the revised manuscript the balance between results and speculative discussion seems ok. The figures are good and clear and the paper should be of interest to others investigating Holocene fluctuations of glaciers and ice caps elsewhere.

N R Golledge 28th Nov 2016

[revised manuscript text omitted]

---

## Author Response (AR3)

**Editor's comments**

**Comments to the Author:**
Decision of editor based on comments of two reviews on revised manuscript and subsequent minor revisions undertaken.

Dear Henning Akesson et al.,
After substantial revisions following rather critical reviews of the initial manuscript, this paper has been reviewed by two further referees which were in general rather positive still but had some suggestions for minor revisions. These minor concerns have now almost entirely been addressed but there is one issue regarding the description of the used mass balance function remaining that after closer inspection I think should be addressed before publication (see details below).This issue can however easily be addressed through technical corrections. I am therefore very happy to inform you that the paper will be accepted after correction of this issue (details below) and I thank the authors for their thorough and careful revisions.

**Minor technical correction:**
Reviewer 1 (of revised version), raised the issue of limited explanation of used mass balance relationship with elevation (in particular decrease in higher accumulation elevations).
Although the authors rewrote parts of the text there I do not think it is in any way clearer and there is no new information in there, rather one line (explaning the varianles in eqn 6) has been shifted a few lines down, making it rather more difficult to follow the explanations.

Further, and I apologies that I have not spotted this earlier, the term 'mass balance gradient' for 'B_Ref' ( and 'B(z,t)') is misleading if not even incorrect (appears on p. 9, line 10 and page 10 line 21). A 'mass balance gradient' refers in glaciology usually to the change of B with elevation (e.g. dB/dz), and I think you mean here rather the 'mass balance function with elevation' or 'mass balance relationship with elevation'.
Further, the additional explanation of this mass balance function is currently in the figure caption (fig 2) but I would think this should be in the methods (in order there to clarify things as suggested by reviewer 1) and maybe one should even add why Giesen and Oerlemans (??) think that the accumulation decreases in high elevations (wind-effects/erosion?).

The editor
Andreas Vieli

**Author's response**
We thank the Editor for suggesting these final corrections. We have clarified the paragraph in Section 3.2.2 describing the mass balance *function* as suggested by the Editor and Reviewer 1 of our revised version. We have also changed "mass balance gradient" to "mass balance function/profile" elsewhere in the manuscript.

Regarding the origin of the observed accumulation decrease at high elevations, we discuss this already in Section 5.2 (p.15, l.19-23 in revised version). We now also mention orographic effects as a possible explanation, as suggested by Reviewer 1.

On behalf of the authors,
Henning Åkesson

[revised manuscript text omitted]